# Common and cell-type specific responses to anti-cancer drugs revealed by high throughput transcript profiling

Mario Niepel [1], Marc Hafner [1], Qiaonan Duan[2], Zichen Wang [2], Evan O. Paull[3], Mirra Chung[1], Xiaodong Lu[4], Joshua M. Stuart[3], Todd R. Golub [4,5,6], Aravind Subramanian[4], Avi Ma'ayan [2] & Peter K. Sorger [1]

More effective use of targeted anti-cancer drugs depends on elucidating the connection between the molecular states induced by drug treatment and the cellular phenotypes controlled by these states, such as cytostasis and death. This is particularly true when mutation of a single gene is inadequate as a predictor of drug response. The current paper describes a data set of ~600 drug cell line pairs collected as part of the NIH LINCS Program (http://www.lincsproject.org/) in which molecular data (reduced dimensionality transcript L1000 profiles) were recorded across dose and time in parallel with phenotypic data on cellular cytostasis and cytotoxicity. We report that transcriptional and phenotypic responses correlate with each other in general, but whereas inhibitors of chaperones and cell cycle kinases induce similar transcriptional changes across cell lines, changes induced by drugs that inhibit intra-cellular signaling kinases are cell-type specific. In some drug/cell line pairs significant changes in transcription are observed without a change in cell growth or survival; analysis of such pairs identifies drug equivalence classes and, in one case, synergistic drug interactions. In this case, synergy involves cell-type specific suppression of an adaptive drug response.

---

[1] HMS LINCS Center, Laboratory of Systems Pharmacology, Department of Systems Biology, Harvard Medical School, Boston, MA 02115, USA. [2] Department of Pharmacological Sciences, BD2K-LINCS Data Coordination and Integration Center, Icahn School of Medicine at Mount Sinai, One Gustave L. Levy Place, Box 1603 New York, NY 10029, USA. [3] UC Santa Cruz Genomics Institute, University of California, Santa Cruz, CA 95064, USA. [4] Broad Institute of MIT and Harvard University, Cambridge, MA 02142, USA. [5] Dana-Farber Cancer Institute, Boston, MA 02115, USA. [6] Howard Hughes Medical Institute, Chevy Chase, MD 20815, USA. Mario Niepel and Marc Hafner contributed equally to this work. Correspondence and requests for materials should be addressed to P.K.S. (email: peter_sorger@hms.harvard.edu)

Understanding why some tumor cells respond to therapy and others do not is essential for advancing precision cancer care. Pre-clinical cell line studies typically investigate the connection between pre-treatment cell state or genotype and drug sensitivity and resistance[1–4]. This approach has proven most effective when oncogenic drivers are themselves targeted by drugs. For example, the presence of EGFR[L858R] (and related mutations) in non-small cell lung cancer (NSLC) is predictive of responsiveness to gefitinib, a drug that binds with high affinity to mutant EFGR[5,6]; the presence of an EML4-ALK fusion protein in NSLC is predictive of responsiveness to crizotinib, which inhibits the ALK4 kinase domain[7]; and the presence of a mutant BRAF[V600E] kinase in melanoma is predictive of responsiveness to the BRAF inhibitors vemurafenib and dabrafenib[8,9]. The Cancer Genome Atlas (TCGA) project and similar efforts are attempting to identify other druggable cancer mutations through molecular profiling of human cancers[10,11], but there is growing evidence that, for many types of tumors and drugs, there exists no simple

genetic predictor of response. For example, genes encoding members of the Akt/PI3K/mTOR pathway are commonly mutated in breast cancer, but the presence of these mutations is a poor predictor of responsiveness to inhibitors of Akt/PI3K/mTOR kinases[12].

A complementary approach, pioneered by the Connectivity Map (CMap)[13] and currently being extended by the NIH LINCS Program, involves collecting molecular data from cells following exposure to drugs and other perturbations and then mining this information for insight into response mechanism. In this paper we report the collection of ~8000 gene expression signatures (in triplicate) from a genetically diverse set of six breast cancer cells exposed to ~100 small molecule drugs by using the low-cost, second generation, CMap technology L1000 transcriptomic profiling (https://clue.io/lincs)[14,15]; in parallel, we measured drug sensitivity at a phenotypic level using growth rate (GR) inhibition[16,17], a method that corrects for the confounding effects of variability in cell division rates, plating density, and media

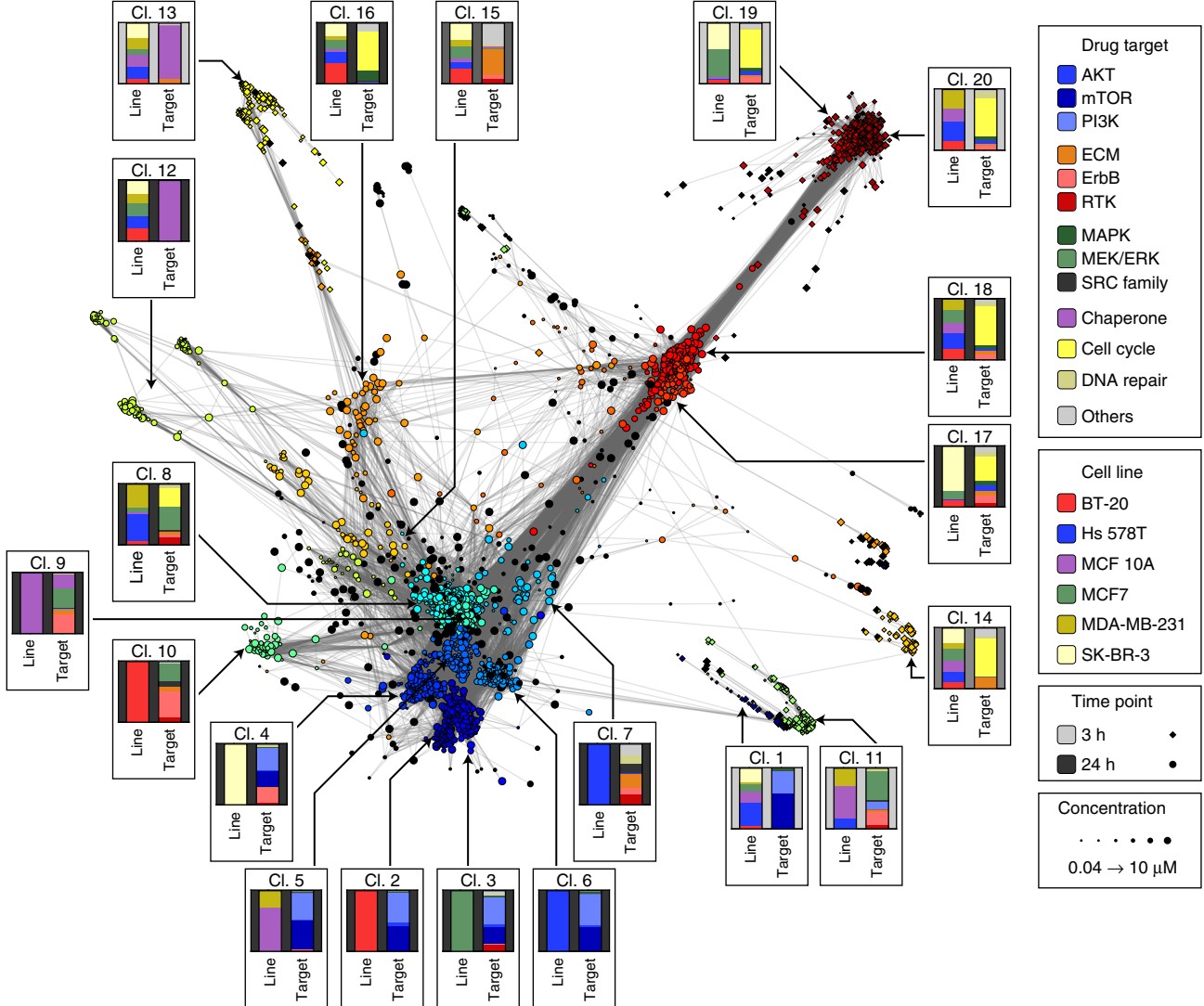

**Fig. 1** Drug responses based on transcriptional signatures can be cell type specific or universal. A network of consistent L1000 signatures (SCS > 1.3) scored using the characteristic direction approach. Each node represents a unique perturbation (a combination of drug, cell line, time point and concentration) and edges are drawn between perturbations having a cosine distance in the lower 2-percentile. Color is based on cluster identity, size denotes concentration, and shape denotes the time point. For each cluster, a box labeled with the cluster number illustrates the distribution of the perturbation by cell line (left bar) and drug target class (right bar). Target classes are assigned based on the nominal targets of a drug and do not consider potential poly-pharmacology. The drugs assigned to each class are listed in Supplementary Data 1. An interactive version of this figure is available at http://amp.pharm.mssm.edu/LJP

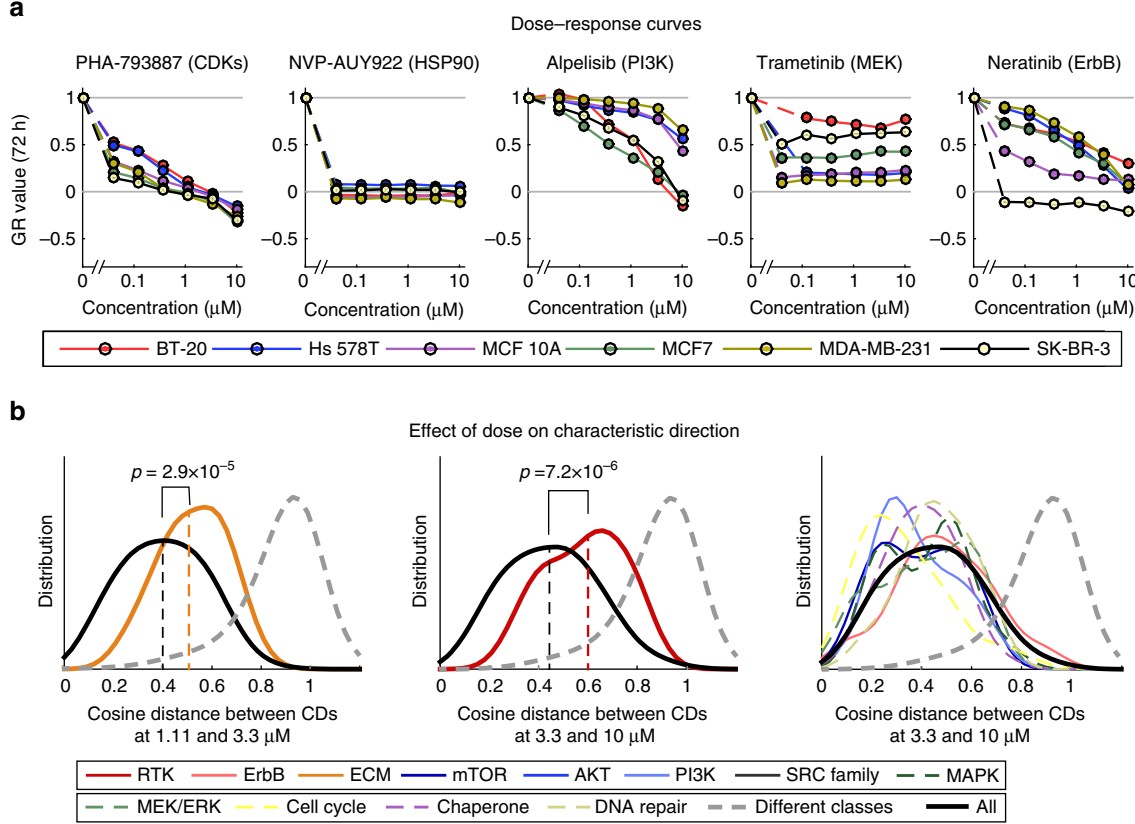

**Fig. 2** Variation of molecular and phenotypic responses by cell line and concentration. **a** Dose–response curves across six cell lines for a subset of drugs that exhibit significant responses at the level of L1000 signature and phenotype. A GR value of one represents growth at the same rate as the untreated control, a value of zero denotes a complete cytostatic response, and a negative value denotes a cytotoxic response. **b** Differences in characteristic directions with increase in drug dose as scored by the cosine distance between the characteristic directions of L1000 signatures. Data for ECM inhibitors between 1.11 and 3.33 μM are shown in yellow (left panel) and receptor tyrosine kinase inhibitors between 3.33 and 10 μM are shown in red (middle panel). For comparison, the data for all other inhibitor classes between 3.33 and 10 μM are shown (right panel). Each curve is a smooth density plot by class of drug target. Each distribution is based on an average of 44, but at least 20, different conditions that generally comprises all cell lines. The black line in all panels shows the distribution of all perturbations, colored lines show the distribution for each class of drug targets as indicated in the legend, and dashed gray line the distribution for angles between drugs of different classes

composition. This data set differs from previous data sets of this type by including transcript data for each drug/cell line pair across dose and time, as well as six-point GR-based dose–response curves based on measurement of viable cell number; GR metrics have higher information content than conventional $IC_{50}$ or $E_{max}$ metrics, and increase the reproducibility of drug-response data[2,16–19].

On the basis of previously published information, we expected that each cell line would exhibit a significant phenotypic response (e.g., cytostasis or death) to only a subset of drugs in our test set[1–4]. The key question was therefore whether cell lines that respond phenotypically to a particular drug do so in a similar way at a molecular level. We found that this was true for some classes of drug, such as inhibitors of cell-cycle kinases: cell lines had very similar sensitivities to these drugs at the phenotypic level and their L1000 signatures were also similar. In contrast, L1000 profiles for drugs such as inhibitors of MAPK or PI3K/Akt signaling, or receptor tyrosine kinases (RTKs) were cell-type specific, even among cell lines in which phenotypic responses were strong. We also identified sets of drug/cell line pairs in which significant changes in transcription were detected without any apparent effect on cell growth. To understand how this might arise we performed a follow-on study showing that BT-20 cells are responsive to PI3K inhibition at a molecular level but that this does not induce cell arrest or death due to the operation of an

adaptive resistance pathway. The adaptive pathway can be blocked by several different drugs whose L1000 signatures co-cluster. Thus, the application of inexpensive, high-throughput transcript profiling combined with cellular phenotypic measurements reveals similarities and differences in responsiveness to anti-cancer drugs depending on genotype and, at least in some cases, can guide the design of effective drug combinations.

## Results

**High-dimensional drug response profiling.** Breast cancer cell lines were selected from the three major subtypes (HER2[amp], HR[+], and triple-negative/TNBC[12]) plus non-malignant MCF 10A cells. A set of 109 investigational and clinically approved drugs enriched in kinase inhibitors was selected to represent "targeted" anti-cancer agents (Supplementary Data 1). Cells were exposed to these drugs at six concentrations over a 250-fold range and samples were collected for L1000 transcriptional profiling at 3 and 24 h. Cell number was measured by imaging at 0 and 72 h and phenotypic responses computed using growth rate inhibition (GR) metrics[16,20,21] (Supplementary Data 2). L1000 profiling measures the levels of 978 "landmark" transcripts in a bead-based Luminex format[14,15]. The expression levels of other genes can be inferred by a computational model trained on transcriptomic data from the Gene Expression Omnibus (see https://clue.io/lincs for

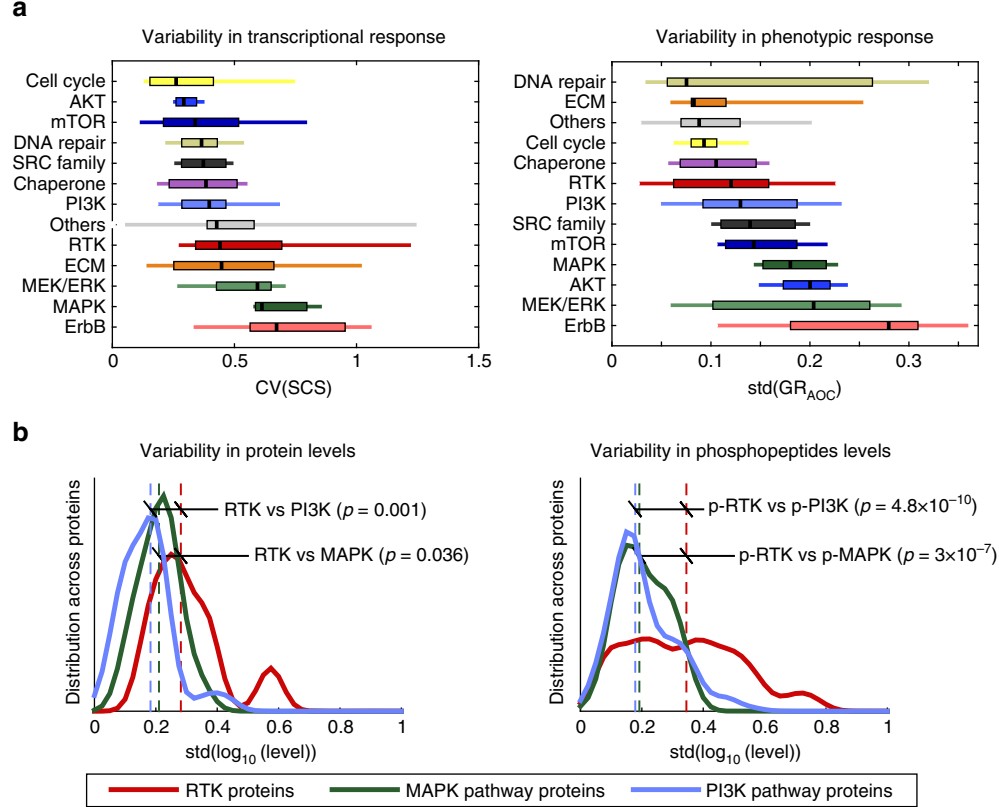

**Fig. 3** Variation in L1000 signature and phenotypic response with drug class. **a** Boxplots of the distributions of the coefficient of variation of the signature consistency score at 24 h, a measure of differences in L1000 signature at all tested doses, by drug target class sorted by the median across all six cell lines (left). Boxplots of the distribution of the standard deviation of the $GR_{AOC}$ (a measure of phenotypic response across all doses for a particular drug) by drug class sorted by the median across all six cell lines (right). Black lines represent the median, boxes the interquartile range, and upper and lower lines the 5 and 95% quantiles. **b** Distribution of standard variation of protein (left) and phospho-peptide (right) levels across six cell lines as measured by shotgun mass spectrometry for proteins in the MAPK and PI3K pathways and of RTKs. On each graph the significance of the difference in distribution between RTK and signaling pathways is indicated. The identities of proteins included in this analysis, as well as complete mass spectrometry profiles, are included in Supplementary Data 4

details). Differential expression analysis was performed on the landmark genes themselves using the "Characteristic Direction" method[22]; a multivariate geometrical approach in which each perturbation is associated with a vector in the 978 dimensions of the L1000 landmark genes. When L1000 profiles differ, characteristic direction vectors point in different directions and the magnitude of the difference can be quantified by the angle between vectors. With large sets of expression profiles, the characteristic direction method outperforms more conventional univariate approaches[22,23]. To account for experimental noise, L1000 signatures were collected in triplicate for each condition and a signature consistency score (SCS) was computed by assessing the degree of alignment of characteristic direction vectors for replicates relative to randomly chosen vectors. We found the SCS to be an effective means for quantifying the reliability of a transcriptional response normalized by background experimental noise (see Methods for details). SCS values correlated to a modest degree with the amplitude of the characteristic direction vector (the effect size; Spearman's $\rho = -0.32$, $p < 10^{-30}$).

Clustering characteristic directions based on the cosine distance (with k = 20 clusters) distinguished responses by drug class, cell line, and time point (Fig. 1). Out of 7825 drug-cell line pairs tested, we included only the subset (2864; 37%) whose L1000 profiles were associated with an SCS > 1.3 value; this served to filter out noisy, low confidence data. Changes in transcription associated with each cluster were interpreted by inferring the full transcriptome (~22,000 genes; https://clue.io/

lincs), averaging the results across all perturbations in the cluster and applying the gene set enrichment tool Enrichr[24] to find terms enriched in up and down regulated genes. To facilitate data exploration, we developed an interactive, on-line tool (http://amp.pharm.mssm.edu/LJP).; in this tool, signatures are tagged with user-selected metadata, drug identity for example, and external transcript signatures are projected onto the network.

Inspection of cell lines and drugs in the most highly populated clusters revealed two patterns of response. In one pattern, multiple cell lines and one or more drugs were found in each cluster. For example, cluster 13 comprised $t = 3$ h signatures from all six cell lines exposed to drugs targeting protein chaperones, and clusters 19 and 20 comprised cell lines exposed to inhibitors of cell cycle kinases or components of the DNA repair machinery (Fig. 1, Supplementary Fig. 1). Three clusters comprised the same set of drugs, but assayed at a later time point ($t = 24$ h; clusters 12 and respectively 17/18); signatures from all six cell lines were once again found in these clusters. Enrichment analysis showed that components of the MAPK and GSK3β signaling cascades were downregulated in the $t = 3$ h (clusters 19/20), whereas cyclin-dependent kinases and genes involved in mitosis were downregulated at $t = 24$ h (clusters 17/18), across all six cell lines. Inhibitors of chaperones and cell cycle kinases were also broadly active at a phenotypic level (Supplementary Fig. 1): for example, the CDK2/5/7 inhibitor PHA-793887 (cluster 18) elicited a near-identical mixed cytostatic/cytotoxic response in all cell lines whereas the HSP 90 inhibitor NVP-AUY922/luminespib (cluster

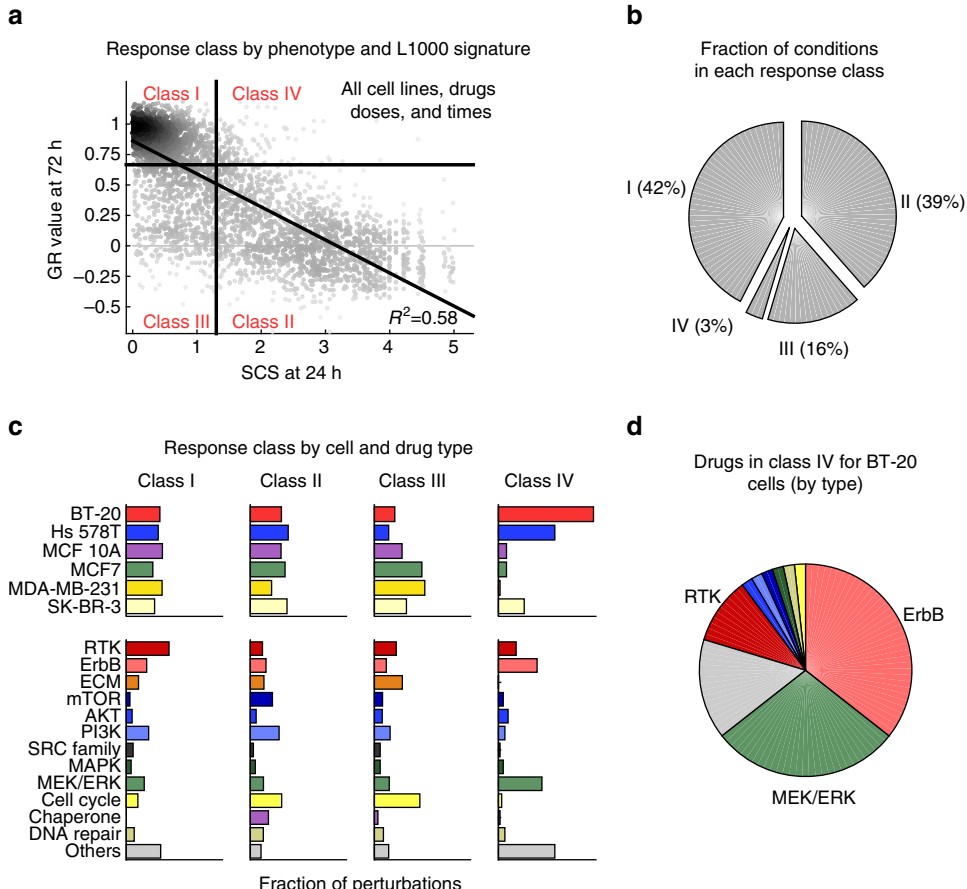

**Fig. 4** Transcriptional responses are generally correlated with phenotypic responses. **a** Scatterplot showing the phenotypic response (GR) and consistency of L1000 responses at 24h (SCS) for all drugs and all cell lines divided into four classes based on a cutoff of GR = 0.66 and SCS = 1.3. Class I: Non-responsive by both measures; Class II: responsive by both measures; Class III: responsive by GR, but not by L1000 SCS; Class IV: responsive based on L1000 SCS, but not by GR. **b** Fraction of perturbations in Class I–IV responses broken down by cell lines (left) and drug target classes (right). **c** Fraction of perturbations in response classes I–IV. **d** Breakdown of Class IV perturbations in BT-20 by drug target classes

12) was uniformly cytostatic (Fig. 2a). We conclude that responses of different cell lines to drugs targeting chaperones, cell cycle kinases, and components of the DNA repair machinery are frequently similar at a molecular level.

A second pattern of response was observed for drugs targeting signaling kinases, such as the PI3K inhibitor alpelisib, MEK inhibitor trametinib, or ErbB inhibitor neratinib. In these cases, clusters frequently comprised (or were dominated by) signatures from one or two cell lines and multiple drugs having closely related targets (clusters 2–11). Phenotypically, responses to such drugs varied with cell line: for alpelisib, this involved differences in potency ($GR_{50}$), whereas for trametinib, maximum effect ($GR_{max}$) varied across cell lines. Even among cell lines that responded similarly at a phenotypic level (for example Hs 578T and MCF 10A treated with alpelisib), the characteristic directions of L1000 signatures clustered the perturbations differently (clusters 5 and 6, respectively). We conclude that responses of cell lines to signal transduction kinase inhibitors differ at a molecular level.

The effects of exposure to drugs at high concentrations might either represent an intensification of responses observed at low concentrations or, alternatively, they might differ qualitatively. Across all drugs and cell lines we observed that with rising dose, the amplitude of the characteristic direction vector weakly but significantly correlated with rising drug concentration (the Spearman correlation factor ranged from 0.13 to 0.32), while the angle of the characteristic direction vector changed only modestly (Supplementary Fig. 2b). Inhibitors of extracellular matrix (ECM) receptors and receptor tyrosine kinases (RTKs) were an exception to this rule: for these drugs, the angle of the characteristic direction vector changed with dose to a greater degree than for other drugs. For example, a significant change in the distribution of cosine distances was observed for ECM receptor inhibitors as concentration increased from 1.11 to 3.3 μM (Fig. 2b, left) or for RTK inhibitors from 3.3 to 10 μM (Fig. 2b, middle) which stands in contrast to the remainder of the drugs (Fig. 2b, right). Moreover, clusters 14, 15, were 17 were highly enriched in drug/cell line pairs corresponding to the highest dose tested (10 μM) for multiple RTK and ECM receptor inhibitors (Supplementary Fig. 1; $p < 0.005$, binominal test). These data suggest that responses to high and low doses of RTK and ECM inhibitors differ at a molecular level and we hypothesize that the selectivity of such drugs is lost at high doses, presumably because they bind to multiple targets.

**Cell type-specific responses of kinase inhibitors**. How do variations in molecular responses and drug-induced phenotypes compare? Answering this question is not straightforward, since the two types of data have different biological meanings and vary in different ways. We settled on a simple comparison in which variation in phenotype was assessed by the standard deviation (SD) of the $GR_{AOC}$ (GR area over the curve), which captures differences in both potency and efficacy (Fig. 3a), and variation in

L1000 signatures was assessed by the coefficient of variation (CV) of the SCS. The statistical power of this analysis is limited by having only six cell lines per drug, but it is informative when drugs are consolidated into classes. Sorting the two types of data

by the median levels of variability (black ticks in Fig. 3a) revealed a consistent trend: inhibitors of cyclin dependent kinases, chaperones, and DNA repair kinases varied little either in L1000 signature or phenotypic responses across cell lines whereas

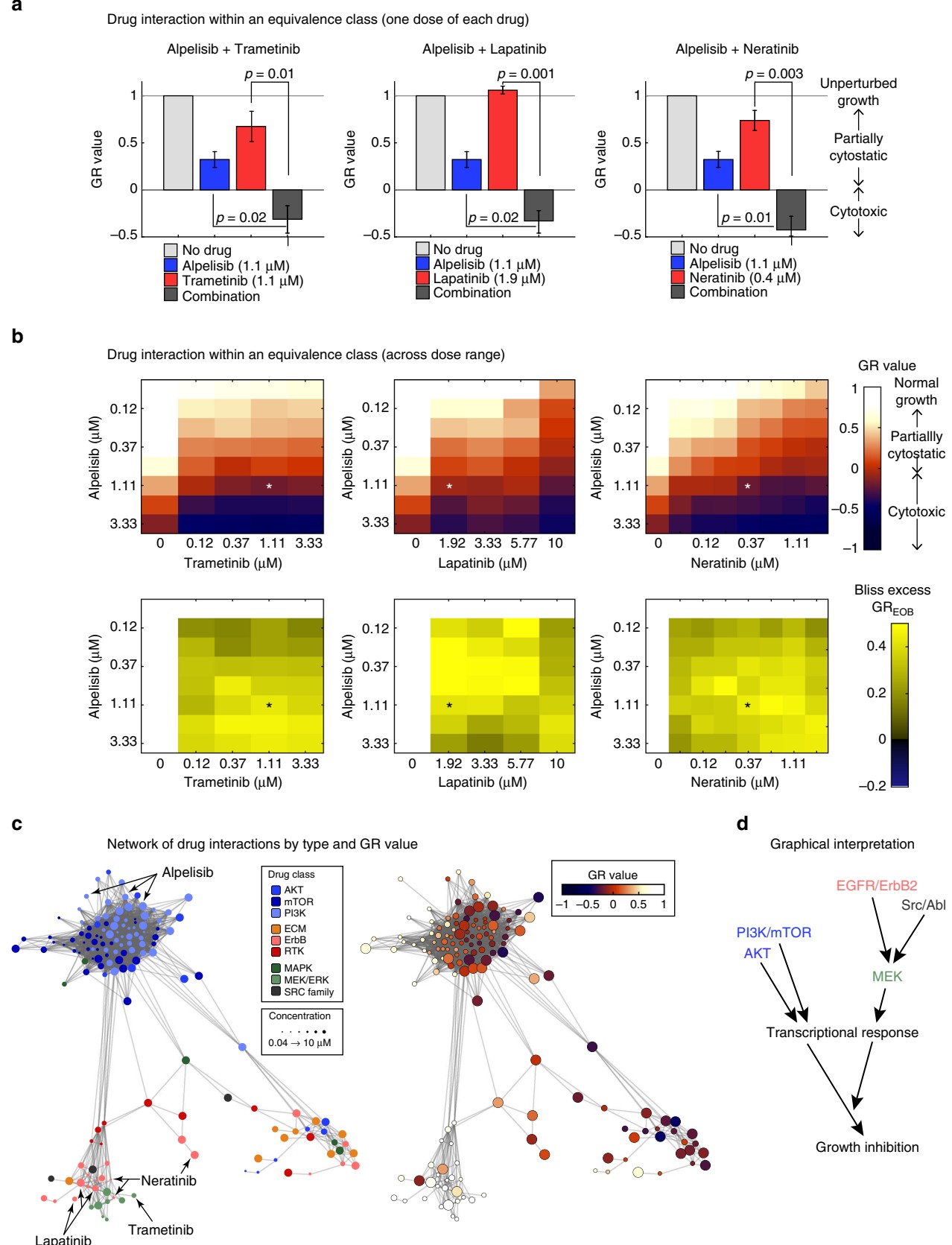

**a** Drug interaction within an equivalence class (one dose of each drug)

**b** Drug interaction within an equivalence class (across dose range)

**c** Network of drug interactions by type and GR value

**d** Graphical interpretation

inhibitors of signal transduction kinases exhibited substantially greater variation. By both measures, ErbB, MAPK, and MEK/ERK inhibitors were the most variable of all (Fig. 3a). This supports our interpretation of Fig. 1: even when cells respond to an inhibitor of a signal transduction kinase at a phenotypic level, the underlying molecular events differ to a greater degree than for other classes of anti-cancer drugs.

We also observed that, by cell type, responsiveness to RTK inhibitors was more variable than for any other class of drugs, but molecular and phenotypic responses were correlated (Spearman's $\rho = -0.65$, $p < 10^{-30}$). Whole-cell shotgun mass spectrometry showed that many intracellular signaling kinases were present at relatively high levels in their active, phosphorylated state in all cell lines, but in contrast, RTKs had variable abundance and degree of phosphorylation across lines (Fig. 3b). Cells sensitive to a specific RTK inhibitor generally expressed the targeted receptor at a higher level relative to other cell lines. In contrast, no such correlation was evident for intracellular signaling kinases, which were expressed and active at relatively constant levels across cell lines. For example, Hs 578T cells expressed high levels of PDGF receptor and were sensitive to nintedanib (and polyselective inhibitor of PDGF/VEGF/FGF receptors), whereas MCF7 cells expressed IGF1R and were sensitive to the IGF1R inhibitors linsitinib and NVP-AEW541 (Supplementary Fig. 2a). High RTK abundance was not sufficient for sensitivity at a phenotypic level, as illustrated by BT-20 cells, which express high ErbB1 levels, but were nonetheless resistant at a phenotypic level to the ErbB inhibitors neratinib and lapatinib. These data suggest that RTK expression is necessary, but not sufficient, for responsiveness to RTK inhibitors; we have previously demonstrated that the same is true of receptor level and ligand response[25]. However, additional studies involving more cell lines will be necessary to determine the statistical significance of this correlation.

**Identification of synergistic drug combinations**. The simplest way to conceptualize the relationship between L1000 and phenotypic data across drugs, concentrations, and cell lines is as a two-by-two matrix with four response classes: (I) cell line/perturbation pairs with no effect by either measure, representing drug resistance; (II) pairs in which responses were observed at both molecular and phenotypic levels, representing drug sensitivity; (III) pairs with measurable phenotypic response, but weak or noisy L1000 signature; and (IV) pairs with substantial changes in L1000 signature, but little or no discernable effect on cell growth. To divide the landscape of SCS and GR values (which are unimodally distributed) into quadrants, we set GR = 0.66 and SCS = 1.3 (Fig. 4). Under these conditions, Class I and II responses were roughly equally probable both for the data set overall and for each cell line and drug class (see Supplementary Data 3 and 4).

Class III responses, representing ~16% of cell line-perturbation pairs, represent cases in which no significant change was detected by L1000 assay, even when significant changes in phenotype were observed. MDA-MB-231 and MCF7 cells were enriched in Class III responses, but when we re-examined the underlying L1000

data, we observed a high level of sample-to-sample variability. We therefore suspect that many Class III responses in our data set arise from poor experimental repeatability (and thus a low SCS score; Fig. 4c and Supplementary Fig. 3a). However, a few drugs were enriched in Class III responses across all cell lines, including cell cycle inhibitors (e.g., barasertib, tozasertib) and extracellular matrix inhibitors (e.g., dasatinib, XMD16–144, PF-431396, PF-562271). These may represent cases in which the L1000 assay does not adequately measure changes in cell state (Fig. 4c and Supplementary Fig. 3b).

Class IV responses represent situations in which a signaling pathway was functional as reflected by a substantial change in L1000 signature, but there was little or no effect of targeting the pathway on cell growth (GR values were high). These responses were the rarest, all, representing only ~3% of drug/cell line pairs. This class was enriched in BT-20 and Hs 578T cell lines and drugs targeting RTKs and the MEK/ERK pathways, which are known to play an important role in the proliferation of breast cancer cells (Fig. 4c). BT-20 cells are known to respond strongly to ErbB ligands such as EGF or heregulin[25] and L1000 data showed that inhibitors of ErbB receptors or MAP kinases (e.g., lapatinib, trametinib) elicited substantial changes in L1000 signatures (Fig. 4d). The effects of lapatinib or trametinib on the growth of BT-20 cells was much weaker than for other cell lines however.

We reasoned that weak phenotypic responses to MAPK or ErbB inhibitors might be due to the presence in BT-20 cells of an activating mutation in the kinase domain of the PI3Kα signaling kinase. Such mutations are common in breast cancer and are known to be powerful oncogenic drivers[10,11,26]. However, BT-20 cells exhibited only a partial cytostatic response to the PI3K inhibitor alpelisib with no cell death observed even at the highest drug concentration tested. We therefore asked whether MAPK signaling might play a role in rescuing BT-20 cells from PI3K inhibitors and vice versa. This is precisely what we observed: when BT-20 cells were treated with a combination of alpelisib and trametinib the effect of the two drugs was synergistic (Fig. 5) as quantified by viable cell number and excess over Bliss independence (GR$_{EOB}$ = 0.56 ± 0.07; $p < 0.05$ by $t$-test; Fig. 5a, left; Supplementary Fig. 4a, left). More dramatically, drugs used in combination elicited a qualitative shift from growth rate reduction to cell death, as evidenced by negative GR values over a wide range of drug concentrations.

The characteristic direction associated with trametinib exposure in BT-20 cells co-clustered with that of the ErbB inhibitors neratinib and lapatinib (Fig. 5b; red and green circles) and the Src kinase inhibitor saracatinib (black circles). In addition, multiple drugs in the PI3K, AKT, and mTOR classes co-clustered with alpelisib. We asked whether drugs that co-cluster by L1000 signature could substitute for each other in drug combinations as judged by their effects on cell viability. We found that this was true both for drugs that shared a characteristic direction with alpelisib and for drugs that shared a characteristic direction with trametinib as quantified both by synergy (GR$_{EOB}$) and a switch from partial growth arrest to cell death (Fig. 5a

**Fig. 5** Drugs falling into Class IV are synergistic with drugs targeting the PI3K pathway. **a, b** GR values for individual drug combinations **a** or over a range of combinations **b** and excess over Bliss scores (bottom) for 72 h exposure of BT-20 cells to combinations of the PI3K inhibitor alpelisib with either the ErbB inhibitors lapatinib (left) or neratinib (middle) or with the MEK inhibitor trametinib (right). Histograms in **a** show the mean of three biological repeats and error bars indicate the standard error of the mean; p-value is based on a t-test. Heatmaps in **b** show data from one out of three biological replicates. **c** Network of significant perturbations (SCS > 1.3) for BT-20 cells. Each node is a unique perturbation (combination of drug, time point, and concentration) and edges are drawn between perturbations with a cosine distance in the lower 5-percentile. Nodes are colored by drugs targeting receptors, the MAPK proteins or components of the PI3K/AKT pathways (left) or the GR value of the response (right). Node size reflects drug concentration. **d** Schematic of the converging effect of drug treatments and illustration of drug equivalence classes in BT-20

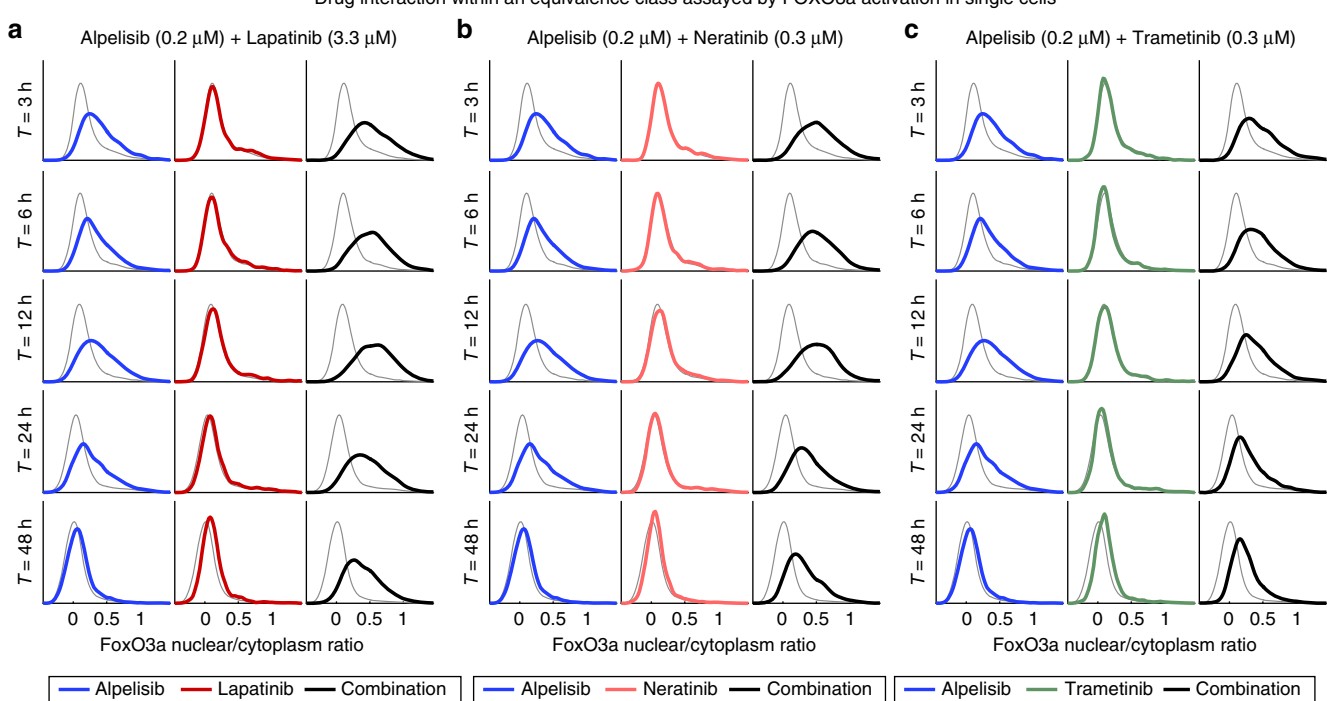

**Fig. 6** Class IV drugs suppress the recovery of FoxO3a signaling. **a–c** Distribution of the nuclear/cytoplasmic ratio FoxO3a as measured by quantitative immunofluorescence microscopy at timepoints indicated. For each distribution, ~2000 cells were analyzed. Cells were treated with alpelisib in combination with either **a** lapatinib, **b** neratinib, or **c** trametinib at the concentrations shown. Gray line show untreated control, blue alpelisib, red the EGFR or MEK inhibitor, and black the combination treatment. Data shown is one out of three biological replicates

middle and right, Supplementary Fig. 4a middle and right, Supplementary Fig. 4b). For example, the $GR_{EOB}$ for alpelisib plus lapatinib or neratinib at the concentrations used in Fig. 5a is 0.64 ± 0.12 and 0.65 ± 0.12 respectively ($p < 0.05$, $t$-test based on GR values). From these data we conclude that co-clustering of characteristic directions highlights compounds that lie within an equivalence class, making it possible to identify multiple combinations of drugs having similar synergistic effects on cell growth (Fig. 5c).

**Mechanisms of drug synergy.** Several recent papers have investigated how to best identify synergistic drug combinations[27–31]. In the case of BT-20 cells, we found that synergy was sometimes associated with combinations of drugs from different response clusters (for example PI3K/AKT inhibitors with EGFR/MAPK inhibitors as discussed above), but such non-overlapping or "orthogonal" transcriptional signatures did not represent the only way to obtain synergy. For example, in Hs 578T cells, the Akt inhibitor A443654 synergized with both the ErbB inhibitor neratinib and cMET/VEGFR inhibitor foretinib, both of which clustered away from A443654 with respect to L1000 signatures and therefore represent examples of potential "synergistic orthogonality." However, the combination of neratinib and foretinib was also synergistic, even though the L1000 signatures associated with these drugs co-clustered and were therefore similar at a molecular level (Supplementary Fig. 5). We conclude that synergy can be observed with combinations of drugs that induce both similar and distinct changes in gene expression, emphasizing our limited understanding of how drug synergy arises.

To investigate the molecular basis of synergy in Class IV responses in BT-20 cells we assayed nuclear translocation of the FoxO3a transcription factor, which (i) is regulated by PI3K, MAPK, and ErbB signaling cascades, (ii) serves as measure of drug response at a single-cell level, and (iii) regulates key aspects of breast cancer physiology such as cell cycle arrest and apoptosis[32–34]. We found that Alpelisib induced nuclear translocation of FoxO3a, as judged by immunofluorescence, reflecting relief of FoxO3a-mediated gene expression. The magnitude of FoxO3a activation by alpelisib was initially high, but then decreased over a 48 h period, presumably due to the previously described phenomenon in breast cancer cells of adaptation to PI3K inhibiton[35]. Lapatinib and neratinib alone had little effect on FoxO3a activity, but when either drug was combined with the PI3K/Akt inhibitor alpelisib, FoxO3a activation was sustained (Fig. 6a, b) and cytostasis enhanced. Trametinib had similar effects as ErbB inhibitors, but the effect on FoxO3a was less strong (Fig. 6c). Thus, inhibition of a time-dependent adaptive response mediated by, or correlated with, FoxO3a translocation, might be the basis of synergy between PI3K and MAPK inhibitors in BT-20 cells. Virtual Inference of Protein activity by Enriched Regulon (VIPER)[36] identified FoxO3a as highly up-regulated following treatment of BT-20 cells with inhibitors of either the PI3K/Akt/mTOR pathway or ErbB receptors, consistent with a role for FoxO3a as a "master regulator"[37] of BT-20 responsiveness to these drugs.

The ErbB inhibitors lapatinib and neratinib inhibited MAPK, but not PI3K, signaling in BT-20 cells (Supplementary Fig. 6a), potentially explaining why ErbB and MAPK inhibitors lie in an equivalence class. Inhibition of MAPK signaling was also transient, which may explain the limited effect of lapatinib and neratinib as single agents. Adaptive responses are often complex at a molecular level[38], but L1000 data suggest that exposure of BT-20 cells to alpelisib results in significant upregulation of the lapatinib targets ErbB1 and ErbB3 (Supplementary Fig. 7), which could be an underlying cause of adaptation. This may also explain why lapatinib and neratinib are substantially more effective than trametinib at preventing recovery of pAKT signaling

(Supplementary Fig. 6b), and thus, why synergy between alpelisib and ErbB inhibitors is so strong. From these data we conclude that discordance between early transcriptional changes and phenotypic responses could prove a generally useful way to identify cases of counter-therapeutic drug adaptation, drugs that can block or overcome such adaptation and phenotypic equivalence across classes of drugs.

## Discussion

In this paper we describe the parallel acquisition of data on drug response at a phenotypic level, as quantified using growth rate inhibition (GR) data[16], and at a molecular level, as quantified by L1000 transcript signatures[14,15], across ~600 drug-cell line pairs and 8000 conditions. The data provide insight into connections between drug-induced changes in intracellular signaling and phenotypes such as cell growth and survival. Because such data have many potential uses, only some of which are explored in this paper, we have made the data available via an on-line browser (http://amp.pharm.mssm.edu/LJP) and direct download (http://www.lincsproject.org/LINCS/data/overview).

On the basis of a large body of prior literature we expected that only a subset of cell lines would respond to a specific drug, but it was not known how molecular changes and phenotypic responses would relate to each other. Overall, we find that the occurrence of drug-induced changes in transcription is generally well correlated with the strength of the phenotypic response. Inhibitors of CDKs or chaperones induced similar changes in intracellular signaling across all cell lines as measured by L1000 signatures, whereas inhibitors of intracellular MAPK and PI3K/Akt/mTOR kinases induced cell-type specific changes. Cell line-specific responses are likely a reflection of differential pathway activity or connectivity, and we speculate that deeper understanding of these differences will improve our ability to use kinase inhibitors in disease. For example, high variability in the effects of PI3K/Akt/mTOR kinase inhibitors in different cells lines might explain the current difficulty in predicting which tumors will respond to these drugs.

Responsiveness to RTK inhibitors was also highly variable across cell lines[1–4], as is expression of RTKs themselves[12]. High expression of a receptor appears to be a good, but not perfect, predictor of responsiveness to drugs targeting that receptor; we have previously found that this is also true of ligand responsiveness[12,25]. When a receptor is present, L1000 signatures elicited by inhibition of a specific RTK were generally similar to signatures elicited by inhibition of either PI3K or MAPK, suggesting preferential pathway engagement downstream of RTK signaling. For example, EGFR/ErbB2 inhibitors clustered with MAPK inhibitors in five out of the six cell lines, but with PI3K/AKT inhibitors in SK-BR-3 cells. The number of cell lines in the current data set is insufficient to determine the statistical significance if such correlations, but future analysis might be directed to larger numbers of cell lines and drugs representative of each cluster in Fig. 1.

Among the most interesting drug/cell line pairs in our data set are those in which a significant change was observed in transcription with no significant effect on cell growth. In the case of the MEK inhibitor trametinib, for example, we found that drug exposure altered transcription in all cell lines, but cell division in only a subset. Thus, MEK is active on its target in all cells, but essential for proliferation only in some. MDA-MB-231 cells, which carry a KRAS mutation, are the most trametinib sensitive, implying possible RAS-MAPK addiction. In contrast, BT-20 cells are the most resistant to MEK inhibition, even though trametinib induces a significant change in transcription. Reasoning that this might reflect the presence in BT-20 cells of an activating PI3KCA mutation, we studied MEK and PI3K/AKT inhibitors in

combination and observed a synergistic effect on cell growth. Synergy between MEK and PI3K/AKT has previously been reported, but we have found this to be true only in a subset of breast cancer cells lines. Synergy in BT-20 cells appears to involve trametinib-dependent inhibition of an adaptive pathway that makes PI3K inhibitors progressively less effective over a 48 h period. Thus, L1000 profiling can uncover relatively subtle mechanisms of drug sensitivity and resistance based on a comparison of transcriptional and phenotypic responses.

Combination therapy is increasingly regarded as essential for treatment of cancer using targeted drugs[27]. The question arises how best to identify combinations that are effective on specific tumors. One approach is to inhibit the same protein or pathway with multiple drugs to achieve better target coverage and functional inactivation of oncogenic signaling. Examples of this strategy include combining pertuzumab and trastuzumab to treat HER2$^{amp}$ metastatic breast cancer[39] and dabrafenib with trametinib to treat BRAF$^{V600E}$ melanoma[40,41]. In other cases, it appears that concurrent inhibition of parallel pathways that drive proliferation or mediate adaptive resistance is the best strategy[38,42–46]. Attempts to distinguish between these possibilities using existing data and sophisticated computational approaches have proven only marginally effective[27,29–31]. Our data show that both "orthogonal" and "non-orthogonal" approaches to drug synergy can be detected in cultured cells. Transcriptional and phenotypic profiling also make it possible to place drugs in equivalence classes based on co-clustering. In the case of BT-20 cells, drugs in an equivalence class can substitute for each other in a combination (lapatinib or neratinib for trametinib in a combination that includes alpelisib). We have not yet studied enough combinations to propose a general rule for the construction of efficacious drug combinations, but the ability of L1000 profiling to identify equivalence classes potentially provides a means for making drug substitutions within a cluster to increase tolerability or effectiveness.

## Methods

**Cell culture and drug response.** Six cell lines (BT-20, Hs 578T, MCF 10A, MCF7, MDA-MB-231, and SK-BR-3) were obtained from the ATCC and grown as published previously[12]. All cells were free of mycoplasma and their identity was verified by short tandem repeat (STR) profiling at the Dana-Farber Cancer Institute[47].

For drug response screening cells were plated at 2000 cells/well (except Hs 578T which was plated at 1000 cells/well) in 384 well plates. After 24 h cells were treated with the indicated doses of small molecule inhibitors obtained from the HMS LINCS drug collection (http://lincs.hms.harvard.edu/). Drugs in this collection are sourced from commercial vendors and subjected to quality control by liquid chromatography–mass spectrometry. Quality control data are available via the HMS LINCS website. For the L1000 assay, supernatant was aspirated 3 and 24 h after drug addition until only 15 µl remained. Cells were then lysed for 30 min at room temperature by adding 30 µl of Buffer TCL (Qiagen). Plates were sealed and stored at −80 °C until processing for L1000 transcriptional profiling. For cell counts, cells were stained with a 1:1000 dilution of Fixable Far Red Dead Cell Stain (Thermo Fisher Scientific) and 2 µM Hoechst 33342 (Thermo Fisher Scientific) for 30 min at room temperature and subsequently fixed with 3% formaldehyde (Sigma Aldrich).

Analysis of single and dual-agent drug response in BT-20 cells involved growing cells as described above followed by direct dispensing of one or two drugs using a D300 Digital Dispenser (Hewlett-Packard). For immunofluorescence experiments, cells were fixed at the indicated time points for 30 min in 3% formaldehyde, permeabilized for 30 min in phosphate buffered saline (PBS) with 0.3% Triton X-100 (Sigma-Aldrich), washed twice in PBS with 0.1% Tween 20 (Sigma-Aldrich; PBS-T), and blocked for 60 min with Odyssey blocking buffer (LI-COR Biosciences). Cells were incubated with antibodies against FoxO3a (Cell Signaling Technologies; 1:200 dilution), phospho-Erk1/2 (Thr202/Tyr204, Cell Signaling Technologies; 1:400 dilution), or phospho-Akt (Ser473, Cell Signaling Technologies; 1:400 dilution) in Odyssey blocking buffer and incubated for 16 h at 4 °C. Cells were then washed three times in PBS-T for 5 min and incubated with Alexa Fluor 647 conjugated donkey anti-rabbit secondary antibody for 60 min at room temperature. Finally, cells were washed two times in PBS-T, once with PBS, and stained for 30 min with whole cell stain (Thermo Fisher Scientific) and Hoechst (Thermo Fisher Scientific), and washed three times in PBS.

All fixed cells were imaged on an Operetta high content imaging system (Perkin Elmer) and analyzed using the Columbus image data storage and analysis system (Perkin Elmer) to determine the number of viable cells.

**Mass spectrometry.** The six cell lines listed above were plated in eight 15 cm dishes at $10 \times 10^6$ cells per dish, except for Hs 578T which was plated at $4 \times 10^6$ cells per dish, and grown for 24 h. Cells were washed twice with cold PBS and scraped off plates in the presence of 1 ml of PBS containing 1:100 Halt Protease and Phosphatase Inhibitor Cocktail (Thermo Fisher Scientific) and then pooled in a one tube per cell line. Cells were pelleted at 900 g, the remaining PBS was removed, and the pellets frozen in liquid nitrogen and stored at −80 °C until lysis. Pellets were lysed in 5 ml of 2% SDS, 150 mM NaCl, 50 mM Tris (pH 8.5), 5 mM DTT, 1:100 Halt Protease and Phosphatase Inhibitor Cocktail. Shotgun proteomic and phospho-proteomic measurements were performed as described[48] and the resulting data is reported in Supplementary Data 5 and 6. For each protein or phospho-peptide, the standard deviation of the log10 of the value is evaluated across all cell lines. Proteins and phospho-peptides are grouped based on their biological function according to the lists in Supplementary Data 7–9.

**L1000 assay.** Extensive information about the L1000 method used in this paper can be found at https://clue.io/sop-L1000.pdf. Briefly, the L1000 assay is performed by amplifying mRNAs from cell lysates by ligation mediated amplification[14]. Probes containing gene-specific sequences are annealed to reverse-transcribed cDNAs, ligated with Taq ligase, amplified by PCR and then hybridized to Luminex beads. The unique fluorescence properties of these beads serve as bar codes. Beads with hybridized PCR products are detected and quantified using a Luminex FLEXMAP 3D reader. 80 transcripts are used to calibrate and normalize data.

**Analysis and clustering of the L1000 data.** For each technical replicate of each drug-induced perturbation, the characteristic direction (CD) was evaluated by comparing its L1000 QNORM vector to vectors for DMSO-treated controls on the same plate[22]. L1000 data is provided at five levels in the data processing pipeline:

- **Level 1:** Raw unprocessed flow cytometry data from Luminex (LXB)
- **Level 2:** Gene expression values per 1000 genes after deconvolution (GEX)
- **Level 3:** Quantile-normalized gene expression profiles of landmark genes and imputed transcripts (QNORM or INF)
- **Level 4:** Gene signatures computed using z-scores relative to the plate population as control (ZSPCINF) or relative to the plate vehicle control (ZSVCINF)
- **Level 5:** Differential gene expression signatures

The normalized values for landmark genes used in the current work correspond in L1000 data sets to "level 3a" data (http://www.lincsproject.org/LINCS/tools/workflows/find-the-best-place-to-obtain-the-lincs-l1000-data). Characteristic direction signatures were calculated per batch. A batch is a group of experimental conditions measured at the same time point and cell-line but on multiple plates as described using the following notation:

- $M$, the number of experimental conditions.
- $N$, the number of control replicates.
- $J$, the number of plates.
- $X_{i,j}$, a vector of length 978 representing the $j^{th}$ replicate of the $i^{th}$ experimental condition. Note that since the replicates of an experimental condition are measured on different plates, $j$ also, typically, denotes the plate.
- $C_{j,k}$, a vector of length 978 representing the $k^{th}$ control replicate on the $j^{th}$ plate.

First, we calculated the CDs for each experimental condition $J$ times, each time using a replicate $X_{i,j}$ and the controls from the same plate to obtain $D_{i,j} = f(C_j, X_{i,j})$, where $f$ is the CD function and $C_j$ is the control matrix $[c_{j,1}, c_{j,2} \ldots c_{j,K}]$ for the plate. Then the final CD, $D_i$, for an experimental condition is:

$$D_i = \frac{\frac{\sum_j D_{i,j}}{J}}{\left| \frac{\sum_j D_{i,j}}{J} \right|}$$

To estimate the significance of the CD, we defined the null hypothesis as the variation of the CDs between technical replicates (same cell line and same treatment) is equal to the variation between the CDs of an equal number of perturbations randomly selected from different cell lines and treatments. If the replicates of a given condition show a significantly smaller variation than the randomly selected perturbations, one can reject the null hypothesis. To calculate a null distribution of appropriately matching characteristic directions we define $S_i$ as the mean of the all possible pair-wise cosine distances between $D_{i,j}$ of the $i^{th}$ experimental condition:

$$S_i = 1 - \frac{\sum_{j'=j+1}^{J} \sum_{j=1}^{J} \left[ \cos\left(D_{i,j}, D_{i,j'}\right) \right]}{\binom{J}{2}}$$

To estimate the null distribution of $S_i$, we randomly drew $J$ number of $D_{i,j}$ from the pool of $M \cdot J$ conditions and calculated their average cosine distance as $S_n$. We repeated the process for 10,000 times to obtain the null empirical distribution.

The Signature Consistency Score (SCS) is the negative log of the one-tail comparison (on the lower end) of $S_i$ with the null distribution $S_n$. The distribution of the $M \cdot J D_{i,j}$ used in the test is not exactly the isotropic distribution, but rather was empirically determined by the aforementioned sampling process because gene expression values are not independent from each other.

**Clustering of CD signatures.** Clustering is based on the cosine distance between the CD signatures with SCS > 1.3 and relies on the algorithm fcm (fuzzy c-means clustering) from MATLAB with an exponent for the membership function matrix of 1.22. Perturbations with a membership of less than 55% in a single cluster were set aside from the clusters (in black in Fig. 1b, right-most column in Supplementary Fig. 1). We performed 101 independent clustering runs and perturbations were assigned to the cluster in which they were found the most often. Using a fuzzy clustering algorithm strongly improve the reproducibility of the results across independent runs. Benchmarking the clustering parameters shows that clustering consistency is high and not dependent on small changes in the parameter values (Supplementary Fig. 8).

A 'consensus signature' is defined for each cluster as the average of the inferred transcriptional signatures within the cluster. MATLAB scripts for all of these calculations are available on https://github.com/sorgerlab/L1000chDir. The results are qualitatively independent of the clustering parameters. Using 16 to 24 clusters results in coarser or finer grouping of perturbations, but not in qualitatively different interpretation of the results with the caveat that setting the number of clusters larger than 20 can result in empty clusters (Supplementary Fig. 8). Network illustrations were created with Cytoscape[49] using the AllegroLayout. Other analyses and figures were constructed using MATLAB.

**Analysis of phenotypic drug response data.** Cell counts were normalized to DMSO-treated controls on the same plate to yield the normalized growth rate inhibition (GR) for each plate (technical replicate) of each cell line at each drug and concentration. Normalized growth rate inhibition was calculated according to the formula: $GR(c) = 2^{\frac{\log_2(x(c)/x_0)}{\log_2(x_{ctrl}/x_0)}} - 1$ where $x(c)$ and $x_{ctrl}$ are the cell counts in drug-treated and, respectively, DMSO-treated control wells, and $x_0$ is the 50%-trimmed mean of the cell count from a day 0 untreated plate grown in parallel until the time of treatment[16,20,21]. Within each experiment, technical replicates (generally three plates) were averaged to yield the mean normalized growth rate inhibition for each cell line, drug, concentration and condition for a given biological replicate.

Synergy of the combination of drugs A and B is evaluated on the GR value using the following formula for the excess over Bliss independence ($EOB_{GR}$):

$$EOB_{GR} = (1 - GR(combination)) - (1 - GR(drugA)) - (1 - GR(drugB))$$
$$+ (1 - GR(drugA))(1 - GR(drugB))$$

**VIPER analysis of the transcriptional signatures.** The VIPER (Virtual Inference of Protein activity by Enriched Regulon) was performed as described in Alvarez et al.[36]. The regulon used for the inference was constructed based on the TCGA data for breast cancer using all genes comprised in the inferred signature as inputs and only transcription factor genes as outputs.

**Data availability.** The main data supporting the findings of this study are available in the Supplementary Data files or by following the links listed in this article. For additional data contact the corresponding author.

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

## Acknowledgements

This work was funded by the NIH LINCS grants U54-HL127365 to P.K.S., U54-HL127624, U54-CA189201 to A.M. and U54-HG006093 to T.R.G. and by a Swiss National Science Foundation fellowship P300P3_147876 to M.H. We thank R.E. and S.G. for help with mass spectrometry, J.W. and N.G. for help with small molecules.

## Author contributions

M.N., M.H. and P.K.S. conceived the study. M.N. and M.H. designed the experiments. M.N., X.L. and M.C. performed experiments. M.H., Q.D., Z.W., A.M., E.O.P. and J.M.S. performed the analysis. Z.W. and A.M. developed the supporting website. T.R.G. and A.S. developed the L1000 assay. M.N., M.H. and P.K.S. wrote the paper and all authors reviewed and approved the final version.

## Additional information

**Competing interests:** A.S. is a founder of Genometry which commercializes L1000 technology. The remaining authors declare no competing financial interests.

