## [Peer Review File · Nature Communications]

Reviewers' comments:

Reviewer #1 (Remarks to the Author):

I think this is a really nice body of work, but there are a few aspects of it that it seems to me aren't as well done as the rest. I don't have simple solutions, but the authors may still care to take some note of my comments and revise accordingly. Everything I say concerns the analyses.

1. First, let me say that I am generally happy with the way the L1000 analyses are handled. I'll summarise, so that the authors can gauge the extent to which I get (or don't get) what they are doing. For each of the 6 cell lines, and some of 109 drugs, and some combinations of these drugs), at 6 concentrations, the expression levels of 978 genes are measured at two time points (3h, 24h). This was done in triplicate, and resulted in 8,000 "gene expression signatures." Since

$6(\text{cell lines}) \times 109(\text{drugs}) \times 6(\text{concns}) \times 2(\text{time points}) \times 3(\text{reps}) > 20,000,$

and this doesn't include drug combinations, something is amiss here. I'm guessing that not everything was done in triplicate, but if that is the case, it raises questions about significance, see below.

All of the gene expression signatures were compared to that of DMSO-only controls, and these comparisons summarised in the high-dimensional vector known as the Characteristic Direction (CD) defined in ref 19. This permits the similarity of two drug-control comparisons to be measured by the cosine of the angle between these CDs, which is extremely convenient, and forms the basis of the clustering of all 8,000 comparisons.

2. But first the significance of the cellular response to the drug is assessed in an apparently novel way, for as far as I could see, it is not from ref 19. The null hypothesis on which empirical p-value calculation is based is not stated, and the calculation not clearly enough described for me to guess it. Further, the two descriptions offered (lines 101-104 and 356-358) do not seem to me entirely consistent. This needs clarifying.

3. The term "amplitude" (of the cellular response) is used twice (lines 103 and 136), but is not defined. Perhaps it is the length of b in equation (2) of ref 19, but we should be told, or mentions of amplitude dropped.

4. It is with reference to amplitude on line 104 that we see the first example of what I see as a frequent misunderstanding of p-values in this paper. The p-value of 1.9×10^{-196} is between p-values and amplitudes of drug-cell line combinations. It is not uncommon for beginners to statistics to interpret p-values as *effect sizes* (smaller p-value goes with bigger effect), but I would have thought this group of authors was better informed than that. A p-value is always calculated under a null hypothesis, and gives no indication of the suitability of alternative hypothesis, where they make sense. A simple web search with the two terms "p-value effect size"

will lead to plenty of reading on this, so I won't labour the point here, though I touch on it below. This is my most serious concern with the paper. As I see it, the authors have chosen to summarize comparisons of the gene expression profiles of drug-treated with control cells, each in 3 replicates, by Characteristic Directions, which is very convenient here, but that comes at a price. Traditional summaries of differential expression include estimated log fold changes (effect sizes) and their associated p-values for the well-defined gene-specific null hypotheses of equal expression level across the two conditions. However, here we do not seem to have effect sizes, unless the unexplained "amplitude" is one such. Instead we only have an empirical p-value, which is poorly explained and possibly spurious, as there are no stated null hypotheses. (Please don't reply that the null is that the mean expression levels of genes in both drug treated and control cells are the same for *all* genes. A p-value for an absurd null hypothesis is absurd.)

5. A further comment here to highlight the naïve use of p-values in this paper. P-values need only be given to 1 significant figure; any more is spurious precision. Presenting $p=1.9 \times 10^{-196}$ tells us that the authors don't know this. Writing "correlated well" to describe a Spearman correlation of 0.32 suggests that the authors think the tiny p-value adds strength to the weak correlation. Even citing the ridiculously small p-value suggests that they don't know that with 8,000 points in a scatter plot, a rank correlation of 0.05 also has a very small p-value.

6. Getting back to the narrative, it seems that the 8,000 CDs are arranged into 20 clusters, and the signatures in any given cluster are used to create a consensus vector. How this is done is not explained. Nor is the way a single consensus vector can be lifted to a pattern of expression for the full transcriptome. Citing ref 14 on line 109 suggests that the answer can be found there, but it can't. So exactly how we get to use Enrichr is not clear. This concludes my summary of the analysis of the L1000 data.

7. I like the basic analysis of the phenotypic data, using the newly developed notion of growth rate inhibition. But what surprised me was the near total absence of any statistical analyses linking the phenotypic and the molecular data. Apart from the plots in Figs 3 and 4, and their counterparts in the Supplementary Figs, which I will mention shortly, the only linking of the two data sets seems to be at the level of informal interpretation, as in lines 113-146. Is this really the best these authors can do, or is a more thorough joint analysis of molecular and phenotypic data to be published elsewhere? Are the examples given the best these data can give us, and were they selected because of a priori interest, or by some algorithm? It would be helpful to know this, to be able to put what is presented into perspective.

8. Let me turn now to the Results. Here I make a few brief comments by line.

Line 98. It would be good to be clear that the differential expression analyses are done with 978 genes, and not with 22,000.

Line 125. Should read “cell lines can respond...”

Lines 136-139. The remarks here seem inconsistent with the content and the caption of Supp Fig 2, where “amplitude” makes no appearance. The content of Figs 2b, 2c and Supp Fig 2 are not quite clear to me. Is each curve a smooth density plot of some collection of cosine distances? If so, roughly how many, covering how many cell lines, drugs?

Line 147-192, comprising pages 7 and 8 and Figs 3 and 4. This seemed to me to be the weakest part of the paper. We have box plots of coefficients of variation of $-\log_{10}(p\text{-values})$, scatter plots of these quantities against GR values, and classes of drug-cell line combinations defined in terms of this quantity. As mentioned earlier, these p-values seem to be surrogates for some measure of overall effect size in a comparison of gene expression levels of drug treated and control cells. As such they will depend on the actual effect sizes, as measured by (say) log fold change for the individual genes, the intrinsic variability in the expression signatures, the technical reliability of the L1000 assay, and the number of replicates. Accordingly, it is hard to believe that the classification of drug-cell line combinations into Class I to IV has any solidity, although the spirit of the classification seems reasonable. It really should be based on something more meaningful than the p-values, which I repeat, have not been properly defined. In my view this needs some hard thinking. The phenotypic data seems conveniently summarized by GR, but summarising a differential gene expression study by one number seems more challenging. If a one number summary is absolutely necessary, then several summaries of the gene-level expression differences could be entertained, but a p-value should not be one of them.

Also on these two pages is a discussion of variability of responses. Again I find this disappointingly superficial. While no-one can argue with the fact that standard deviations (SDs) are legitimate measures of variability, they are hardly biologically meaningful, depending as they do on the assay. Also, empirical SDs have variances that depend strongly on their means, and so it is generally advised that any statistical analysis using SDs should be based on their logs. Further, it is not clear how many observations went into each estimated standard deviation. The caption to Fig 3b suggests 6, and if this is true, then these are very noisy SDs indeed. Finally, the provision of p-values for comparisons of collections of SDs, is an example of what I call p-value overkill. No null hypothesis is stated, but if it were, it would be that the distribution of some class of SDs is identical to that of some other class. Spelling it out like this highlights just how far this analysis is from anything that has any biological meaning. My reading of this aspect of the analysis is that the authors looked informally at the data, saw some trends, and decided to make their exploratory observations look scientific by decorating them with p-values. If so, this is in the best tradition of data snooping. If not, and the authors claim to be testing a hypothesis stated in advance of collecting the data, then that should be stated clearly and its scientific meaning explained.

Line 152. Here we meet GR_{AUC} . In ref 16 by four of the present authors, it is explained that GR_{AUC} has been superceded by GR_{AOC} . Is this a typo, and AUC here should be AOC , or is the old AUC notion back in favour?

Line 194. Regarding the use of EOB, has there been any rationale given for the use of this particular definition for GR? Multiplicative definitions of independence are notoriously dependent on the “scale”, i.e. the nature of the measure. The use and misuse of measures of synergy has a large literature, but to my knowledge, none of it mentions GR.

Line 442. Fig. 6. The caption does not tell us what the grey lines are, although we might infer that they represent untreated control cells. Why are there replicate grey lines but not for the drug treated cells?

Lines 471 and 475. First, note that the captions of Supp Figs 5 and 6 have been switched. Second, although “untreated control cells” are mentioned in the caption to Supp. Fig 6(which should be 5), their color (grey) is not mentioned.

Supp. Fig 6 seems rather poorly drawn.

Reviewer #2 (Remarks to the Author):

A. Summary of the key results

The authors present a large data set of 6 breast cancer cell lines exposed to 109 drugs where the transcriptional and phenotypic responses were recorded at multiple time points. They use these data to cluster the cell-line-drug-time point entities and then draw a number of conclusions. It is interesting that the authors can identify drugs that have the same effect across cell lines, and drugs that have cell line specific effects. This might have been expected, but it has not been shown on this scale and is thus an interesting find. The most interesting results for follow-up are the cases where the gene expression and phenotypic response do not correlate (Classes III and IV). For Class III, the authors provide as reason that the gene expression assay is not sufficiently sensitive. Indeed the cell line with the poorest quality has the highest number of Class III measurements. The authors use Class IV to identify cases of synergy, and provide some follow up study for at least one of these cases. The dataset is very interesting, and it is great that the authors share it freely, as it can be a very useful resource as temporal responses of cell lines to drug perturbations will likely yield a better understanding of (combinatorial) drug response.

B. Originality and interest: if not novel, please give references

The data set is interesting, and showing transcriptional and phenotypic responses at this level is novel. Identification of drug synergies is also very important, However, the authors are not very clear on how this dataset can be exploited to identify potential synergistic drug partners for a given cell line.

C. Data & methodology: validity of approach, quality of data, quality of presentation

Using the L1000 to record transcriptional responses seems like a reasonable approach. At the least it brings such large scale transcriptional profiling experiments within reach. Whether the L1000 generates reproducible data and whether the L1000 readout is a good transcriptional readout for drug response is another question - see Comment 6.1 below.

The use of the fuzzy clustering approach is reasonable, but we also have some suggestions for improvements here, especially regarding parameter settings. See Comment 6.3 below.

Regarding the presentation: The paper is hard to read. Specifically the methodology is hard to follow if one is not completely familiar with the characteristic direction approach, for example. The manuscript will greatly benefit from a small description of the intuition behind this approach. There also seems to be a disconnect between the first half of the paper where the data is presented, described and high-level conclusions are drawn and the second half where the authors explore synergistic relationships. More clear descriptions of how one gets from the clustering to the identification of synergies will be very useful. Specifically, it is unclear whether the synergy described in BT20 (MAPK and PI3K) is extracted from the data or from the literature.

D. Appropriate use of statistics and treatment of uncertainties

In general, these are satisfactory, but we have specific comments regarding the excess over Bliss, p-values of the replicates and the reproducibility of the L1000 measurements.

E. Conclusions: robustness, validity, reliability

In Section 6 we make several suggestions to improve the robustness and validity of the conclusions.

F. Suggested improvements: experiments, data for possible revision

6.1 A technical description of the L1000 measurement system is lacking, and the authors refer to another publication which is in preparation (Line 343: "A full description of the L1000 method is being prepared for publication ..."). While we appreciate that all results can not be presented in a single manuscript, it is appropriate to share some of the data relevant to this work from the L1000 manuscript. Specifically it is relevant to this work to show 1) how well the L1000 captures transcriptional profiles (how much we miss of the transcriptional responses by not doing a full transcriptome, and how L1000 compares to conventional transcriptional profiling on the genes measured by L1000) and 2) what is the reproducibility of the L1000 platform. There is some mention of replicate measurements and how their characteristic directions compare to random perturbations, but more detail is required (see Comment 6.2).

6.2 Similarly, for the data processing, there is insufficient information; for example, what is the "QNORM level 3" data? (p 15 at bottom.) How was this calculated from the intensity measurements? For the calculation of the significance of response in gene expression (p 16 at the top) a more detailed description should be given of how the empirical p-values were calculated (preferably with equations). For example, with how many random pairs were used to construct the null model?

6.3. Regarding the clustering approach:

6.3.1 What was the reason for using fuzzy clustering? Did the fuzzy clustering provide a benefit over other clustering methods? It appears that in the end, the drug/cell line pairs are assigned to a single cluster again. Specifically, the authors state on p 16, L363: "...perturbations were assigned to the cluster in which they were found most often". How does this relate to fuzzy clustering. Isn't the point of fuzzy clustering that cluster membership is not discrete and that a perturbation can belong to multiple clusters?

6.3.2 For the clustering analysis, a stability analysis is required before strong conclusions can be drawn from it. How does the clustering hold up under bootstrapping, for example? The authors should also provide a justification for the choice of $k=20$ and the fuzziness parameter of 1.22.

7. For the synergy prediction, the specific case of synergy that is studied is one that is already well-known. Indeed the authors are aware of this, as they mention that "breast cancer cells typically requires both MAPK and PI3K signaling". Thus, the finding that BT20 cells depend on both MAPK and PI3K signaling is not novel. The authors should mention (also in the abstract) that they uncover a known synergy. As such it does still provide a validation of the strategy.

8. It was also not entirely clear how the authors arrived at this synergy prediction; was that entirely data-driven? It would be interesting if the authors could provide a list of the other cases of potential synergy that can be identified from the data in the same manner. Are there any novel predictions? It appears one other drug combination was tested and validated as well (Foretinib + A443654), but this is not further discussed.

9. For the synergy measurements, are there indeed no replicates? Also, how strong is a 'Bliss excess' of $\sim 0.2-0.4$? The significance of synergy is certainly dependent on the noise in the measurements. For example, the uncertainty in the single drug response measurements will feed through into the computation of the expected Bliss score, i.e. this score has a certain confidence interval. If the excess score is within this interval one can not call it significant. The authors should provide some estimate of the uncertainty on the Bliss score and attach a p-value to the excess score.

10. The title of the section: "EGFR/MAPK inhibitors drive synergy by blocking drug adaptation" is too strong a statement. Also in the abstract the authors states that "we [...] show that it blocks adaptive drug resistance". In our opinion this is not supported by the evidence. While the authors show that the combined inhibition of MAPK and EGFR/ERBB2 leads to different dynamics in FoxO3A, it is not shown that this effect drives the synergy. Correlation does not imply causation.

Intervention studies should be done to be able to prove that this mechanism is responsible/underlying the synergy. Additionally, the synergistic effect (in altering FoxO3A dynamics) is stronger between EGFR and PI3K inhibition, compared to MAPK and PI3K inhibition, hinting that more than one effect is at play here.

11. It is unclear what the benefit is of inferring gene expression of unobserved genes from the expression of the landmark genes, and for which analyses this is used. Only for the gene set enrichments?

12. Identification of FoxO3a is based on a method (VIPER) that is not published yet, and no description is given of this method. Was FoxO3a the only hit or the top hit. If not, how was it selected?

13. page 7 line 157. Cells sensitive to a specific inhibitor generally expressed the targeted receptor. How general is this. It is only shown with selected examples. This point should be supported in a more unbiased way?

G. References: appropriate credit to previous work?

Yes

H. Clarity and context: lucidity of abstract/summary, appropriateness of abstract, introduction and conclusions

See comments under 3 above regarding the presentation.

Reviewer #3 (Remarks to the Author):

Niepel, Hafner et al. present a large-scale interrogation of breast cancer cell lines treated with anti-cancer drugs by growth assessment and transcriptomic profiling. The authors find that these measures are overall well correlated but that certain drug classes induce more cell line specific responses than others. The manuscript also proposes this method as a way to identify synergistic drug combinations.

In general, the experiments are technically well performed, the data analysis appears sound and the manuscript is well written. I appreciate the parallel presentation of the data in an interactive website format.

Whereas the study is conceptually not novel and several similar drug profiling approaches have been performed after the Connectivity Map (Lamb et al, 2006) was published, the authors employ an interesting and cost-effective method for transcriptional profiling to generate large datasets. However, the details and benchmarking experiments of this method are not part of this particular manuscript, making it difficult to judge its performance and diminishing the impact of the manuscript.

The strength of the manuscript lies in the large dataset and its analysis that connects drug response to transcriptional changes across time, dose and cell lines. This really is rather unique dataset and worth publishing. However, I struggle to identify the key novel findings that would make it a broad interest paper. Those highlighted by the authors seem somewhat expected and trivial to me. I.e., inhibition of well-conserved processes that are not differentially mutated in cancer cell lines yield similar profiles, whereas targeting highly context dependent and cancer driving processes (e.g., signaling pathways, RTKs) are much more variable. The synergy aspect of the paper is interesting but not yet worked out to a sufficient depth to make a case that this is really an effective approach to identify clinically meaningful synergies.

Other comments:

- The mention of 8000 combinations is misleading as it suggests 8000 unique drug-cell line combinations.
- To call a cell viability measurement "Cellular phenotype profiling" seems a stretch
- To me, the described increase in amplitude with increasing concentration is not obvious in Fig. S2.
- Similarly, the claim that 10uM signatures of RTK and ECM inhibitors co-cluster across all cell lines is not easily visible from Figs. 1 and S1.
- The authors should include a plot showing the data described for cell type specific responses of kinase inhibitors (lines 158-166) and cytostasis of alpelisib + ErbB inhibitors (line 223).
- At what time point were the synergy experiments measured? This seems an important parameter to me, given that these are class IV drugs, i.e. showing a transcriptional but no growth response.
- Do the grey lines in Figs. 6 and S5 represent untreated or DMSO controls? If yes, these should be labeled as such; if not, these need to be included. Maybe representative IF images could help illustrate these findings. Further, the inhibition of MAPK signaling in Fig. S5a is not obvious to me.
- Fig. 1 is difficult to read. Maybe larger or more different symbols for time points and circling/grouping related clusters would help.
- In general, whenever individual perturbations/drugs are mentioned in the text, these should be highlighted in the corresponding figures to allow the reader to follow the line of argumentation (e.g. barasertib etc. in S3, lapatinib etc. in 4c and 5b).
- Right panel in Fig. 2a (Alk inhibitor), AZD6244 synergies (Fig. S4a) and Suppl. Data 2 are not referred to in text.
- Fig. S1: The authors claim that chaperone and cell cycle inhibitors are active across all cell lines. This is not directly visible from the plots presented. The authors could plot e.g. the fraction of cell lines by drug class to illustrate this better. They should also show cluster 21 in the last panel.
- Lines 105ff.: Mentioning the cosine distance in this paragraph would facilitate understanding.
- Line 144: In my opinion, there are more than 6 clusters containing signatures for multiple cell lines.
- Lines 129/130: Clusters 5 and 8 comprise more than one cell line.
- As the VIPER method is not yet published, a slightly more detailed description would be helpful.
- Legends for Figs. S5 and S6 are swapped and somewhat unclear (chDir? which cell line is used?).
- What do DOX concentration and batimastat treatment refer to (line 369)?
- Some authors are not mentioned in the author contribution/ conflict of interest section.
- Missing labels in figures: y axis in S2, labels for classes in S3, legend for grey lines in 6 and S5
- Typos and unclear phrasing: line 79 significant, line 83 which combination?, line 110 s with, line 126 response (wrong word), line 167 of missing, line 263 are similar, line 280 effective, line 314 involved, line 324 room temperature, line 402 do; Fig. S1 107 M and y-axis label (distribution)

Response to reviewers comments for NCOMMS-16-14817

Reviewer #1 (Remarks to the Author):

1.1 First, let me say that I am generally happy with the way the L1000 analyses are handled. I'll summarise, so that the authors can gauge the extent to which I get (or don't get) what they are doing. For each of the 6 cell lines, and some of 109 drugs, and some combinations of these drugs, at 6 concentrations, the expression levels of 978 genes are measured at two time points (3h, 24h). This was done in triplicate, and resulted in 8,000 "gene expression signatures." Since

$6(\text{cell lines}) \times 109(\text{drugs}) \times 6(\text{concns}) \times 2(\text{time points}) \times 3(\text{reps}) > 20,000$, and this doesn't include drug combinations, something is amiss here. I'm guessing that not everything was done in triplicate, but if that is the case, it raises questions about significance, see below.

R1.1 By way of clarification: we apologize for the confusion – the number of L1000 signatures as reported in the paper is following merger of technical triplicates. We did not collect L1000 signatures when looking at the effects of drug combinations. Thus, the number of signatures is:

$$6(\text{cell lines}) \times 109(\text{drugs}) \times 6(\text{concs}) \times 2(\text{time points}) = 7848 \text{ (times three replicates)}$$

1.2 But first the significance of the cellular response to the drug is assessed in an apparently novel way, for as far as I could see, it is not from ref 19. The null hypothesis on which empirical p-value calculation is based is not stated, and the calculation not clearly enough described for me to guess it. Further, the two descriptions offered (lines 101-104 and 356-358) do not seem to me entirely consistent. This needs clarifying.

R1.2 As the reviewer pointed out, we had to develop novel way to analyze the L1000 data as there is no established approach for analyzing such high-dimensional signatures. We now clearly describe a new set of tools and metrics for evaluating signatures; this is reflected in changes throughout the text.

We agree with the reviewer that assessing the 'significance of cellular response' by p-values is problematic and our presentation was confusing (please also see cover letter on this point). We now define significance as the 'signature consistency score' (SCS) – namely the comparison of the characteristic direction of two replicates to two randomly picked perturbations (the value is the negative log₁₀ of the fraction of random CDs with an angle smaller than the angle between technical replicates) and we modified our analysis, text, and figures to reflect this change. We have revised the methods section to describe how SCS is calculated and hope that this change addresses the reviewers concern. Despite the change in methodology, the overall conclusions from our analysis remain unchanged.

1.3 The term "amplitude" (of the cellular response) is used twice (lines 103 and 136), but is not defined. Perhaps it is the length of *b* in equation (2) of ref 19, but we should be told, or mentions of amplitude dropped.

R1.3 By way of clarification: we now explicitly define 'amplitude' in the methods section. Briefly, amplitude is defined as the average (across replicates) of the length for the projection of an L1000 vector in the characteristic direction.

1.4 It is with reference to amplitude on line 104 that we see the first example of what I see as a frequent misunderstanding of p-values in this paper. The p-value of 1.9×10^{-196} is between p-values and amplitudes of drug-cell line combinations. It is not uncommon for beginners to statistics to interpret p-values as effect sizes (smaller p-value goes with bigger effect), but I would have thought this group of authors was better informed than that. A p-value is always calculated under a null hypothesis, and gives no indication of the suitability of alternative hypothesis, where they make sense. A simple web search with the two terms "p-value effect size" will lead to plenty of reading on this, so I won't labour the point here, though I touch on it below. This is my most serious concern with the paper. As I see it, the authors have chosen to summarize comparisons of the gene expression profiles of drug- treated with control cells, each in 3 replicates, by Characteristic Directions, which is very

convenient here, but that comes at a price. Traditional summaries of differential expression include estimated log fold changes (effect sizes) and their associated p-values for the well-defined gene-specific null hypotheses of equal expression level across the two conditions. However, here we do not seem to have effect sizes, unless the unexplained “amplitude” is one such. Instead we only have an empirical p-value, which is poorly explained and possibly spurious, as there are no stated null hypotheses. (Please don’t reply that the null is that the mean expression levels of genes in both drug treated and control cells are the same for all genes. A p-value for an absurd null hypothesis is absurd.)

R1.4 We have substantially changed the description of the newly-developed set of tools and metrics for evaluating signatures; this is reflected in changes throughout the text, but without a significant impact on our overall conclusions.

Of course, we agree with the reviewer on this point, and fully appreciate the potential for confusing p-values with effect size; we thank the reviewer for pointing this out. After some consideration of the strengths and potential weaknesses of reduced dimensional L1000 signatures, we have concluded that amplitude (which we now define explicitly— see R1.3) is the most appropriate measure of effect size. The newly defined term ‘signature consistency score’ (SCS – see R1.2) captures the reproducibility of the L1000 signature. It takes the place of the ill-defined empirical p-value. For reasons related to the variability of the L1000 assay across assays plates and biological replicates, we believe that the SCS is the best means available for characterizing signatures that are computed using the characteristic direction method. We have made changes at multiple points in the results and method sections to clarify this point.

1.5. A further comment here to highlight the naïve use of p-values in this paper. P-values need tells us that the authors don’t know this. Writing “correlated well” to describe a Spearman correlation of 0.32 suggests that the authors think the tiny p-value adds strength to the weak correlation. Even citing the ridiculously small p-value suggests that they don’t know that with 8,000 points in a scatter plot, a rank correlation of 0.05 also has a very small p-value.

R.1 5 By way of clarification: these are indeed issues of which we are aware. We do not in fact believe that low p-values in this case make a Spearman correlation of 0.32 any more biologically informative; the use of the term “correlated well” is therefore overstated for $r=0.32$ and we have re-written the text accordingly. This change does not weaken any important conclusions in the paper.

1.6. Getting back to the narrative, it seems that the 8,000 CDs are arranged into 20 clusters, and the signatures in any given cluster are used to create a consensus vector. How this is done is not explained. Nor is the way a single consensus vector can be lifted to a pattern of expression for the full transcriptome. Citing ref 14 on line 109 suggests that the answer can be found there, but it can’t. So exactly how we get to use Enrichr is not clear. This concludes my summary of the analysis of the L1000 data.

R1.6 By way of clarification: We apologize for this oversight. We now describe (in both the main text and in the methods section) how we create consensus expression signatures for each of the 20 drug response clusters, which simply reflects the average of the inferred transcriptional signatures of the perturbations comprised in the cluster.

1.7 I like the basic analysis of the phenotypic data, using the newly developed notion of growth rate inhibition. But what surprised me was the near total absence of any statistical analyses linking the phenotypic and the molecular data. Apart from the plots in Figs 3 and 4, and their counterparts in the Supplementary Figs, which I will mention shortly, the only linking of the two data sets seems to be at the level of informal interpretation, as in lines 113-146. Is this really the best these authors can do, or is a more thorough joint analysis of molecular and phenotypic data to be published elsewhere? Are the examples given the best these data can give us, and were they selected because of a priori interest, or by some algorithm? It would be helpful to know this, to be able to put what is presented into perspective.

R1.7 We have performed additional analysis and substantially changed the text and figures in response to this comment. Our response to this point is further elaborated in R1.11 to R1.13.

As we point out in the cover letter, the dataset in this paper is unusual in comprising a systematic set of measurements of transcriptional state (albeit with reduced dimensionality) and cell phenotype (cytostasis or cytotoxicity). Our goal was to understand how such measures relate to each other and to better understand the source of variation. However, off-the-shelf tools are not readily available for this sort of analysis and we have therefore combined some systematic investigation with attempts to answer specific biological questions.

To begin to address the concern that more statistical analysis is warranted, we have added a supplemental figure that breaks down the correlation of transcriptional and molecular responses by cell line and drug class to further explore regularities and outliers in the data set. We appreciate that this does not fully address the reviewers concern. It is simply not yet known how the molecular changes detected by transcript profiling (or other multiplex molecular methods) should be related in a systematic manner to phenotypic responses. The molecular changes that give rise to reduced rates of cell proliferation, or increased rates of apoptosis, in drug-treated cells are not known beyond invoking rather general concepts such as “oncogene addiction.”

Our paper provides a possible explanation for this deficiency, namely that the signatures associated with inhibition of signal transduction kinases are in many cases cell type specific (hence, the clustering by cell line that we observe in Figure 1). Attempts by many groups (ours included) to develop a pan-cell-type models of the effects of MEK/ERK, PI3K/AKT, and RTK inhibitors on cancer cells are likely to have foundered on this diversity of response: at least at this phenomenological level, there may not exist a unified mechanism to explain the effects of a particular drug on multiple cell types.

To address this issue, we are now attempting to complement the large-scale survey presentation in this paper with a detailed drill-down of specific drug examples. The current paper presents several such examples (in BT-20 cells) in which it was possible to use VIPER to identify potentially causal connections between drug-induced changes in transcription and the activities of upstream factors. Existing knowledge about these factors allowed us to develop a story about “adaptive” drug responses, drug combinations and drug substitution. A fair amount of manual effort and literature searching was required to derive this result. Thus far, we have not managed to apply such an approach in a systematic fashion across the entire dataset (among other things, further development of inference algorithms such as VIPER is necessary), and we are not holding back additional insights at the level of the entire dataset. However, we have put extensive effort into an on-line data analysis tool (<http://amp.pharm.mssm.edu/LJP/>) that performs many different types of gene set enrichment analysis.

Rather than hold back the data, we believe that it is appropriate to publish what we have achieved and allow others the opportunity to improve on our approach. We are motivated by the fact that the NIH LINCS program that funds this work mandates the rapid release of results and methods to the community as a means to promote crowdsourced solutions (by analogy with datasets such as ENCODE and TCGA). To this end, we are making all aspects of the current work freely available and we created the interactive website to display and disseminate the data and to stimulate interest in the development of new tools for linking transcriptional and phenotypic measures of drug response.

1.8 Line 98. *It would be good to be clear that the differential expression analyses are done with 978 genes, and not with 22,000.*

R1.8 This is an excellent point: we now make it explicit in the text when we are working with the ~978 measured genes or with the inferred profile of ~20,000 genes.

1.9 Line 125. *Should read “cell lines can respond...”*

R1.9 Thanks – we have corrected the text accordingly.

1.10 Lines 136-139. The remarks here seem inconsistent with the content and the caption of Supp Fig 2, where “amplitude” makes no appearance. The content of Figs 2b, 2c and Supp Fig 2 are not quite clear to me. Is each curve a smooth density plot of some collection of cosine distances? If so, roughly how many, covering how many cell lines, drugs?

R1.10 We corrected the text to be consistent with the content and caption of Figure S2. We also explain Figures 2b, 2c, and S2 more clearly and include the Pearson’s correlation coefficient between drug dose and amplitude in the text to indicate their degree of correlation.

1.11 Line 147-192, comprising pages 7 and 8 and Figs 3 and 4. This seemed to me to be the weakest part of the paper. We have box plots of coefficients of variation of $-\log(p\text{-values})$, scatter plots of these quantities against GR values, and classes of drug-cell line combinations defined in terms of this quantity. As mentioned earlier, these p-values seem to be surrogates for some measure of overall effect size in a comparison of gene expression levels of drug treated and control cells. As such they will depend on the actual effect sizes, as measured by (say) log fold change for the individual genes, the intrinsic variability in the expression signatures, the technical reliability of the L1000 assay, and the number of replicates. Accordingly, it is hard to believe that the classification of drug-cell line combinations into Class I to IV has any solidity, although the spirit of the classification seems reasonable. It really should be based on something more meaningful than the p-values, which I repeat, have not been properly defined. In my view this needs some hard thinking. The phenotypic data seems conveniently summarized by GR, but summarising a differential gene expression study by one number seems more challenging. If a one number summary is absolutely necessary, then several summaries of the gene-level expression differences could be entertained, but a p-value should not be one of them.

R1.11 We have performed additional analysis and substantially changed the text and figures in response to this comment (Please also see response R1.7). This includes looking more carefully into cell lines and drugs that fall into the responders or non-responders, and further inspection of outliers; these results are incorporated as additional panels into Figure 4 and discussed in more detail in the text. As discussed above we have replaced ill-defined p-values with a signature consistency score (SCS) which we explain in the methods section and the text.

It is worth noting again that to our knowledge the current dataset is the first to compare systematically the transcriptional and phenotypic measures of drug response across many drug/cell lines pairs. Performing such an analysis is not straightforward because phenotypic and transcriptional data are different in type, dynamic range, noise characteristics etc. We therefore approached the issue from the perspective of some simple biological questions rather than a statistical model.

As a first pass at the data we wanted to determine how variable each measurement was (e.g. do all cells respond similarly to the same drug?) and probe the mechanism of non-responsiveness (e.g. if cells are non-responsive at a phenotypic level are they also non-responsive at a transcriptional level?). Simple inspection (e.g Figure 1) suggested that these might be interesting questions since responses to some drug appeared uniform and other quite variable. We do not claim in Figs. 3-6 to have done anything more sophisticated.

In this context, our 2 x 2 classification of response data was aimed at understanding why molecular and phenotypic response were not correlated to the same extent across cell lines and drugs. This naïve but, intuitively appealing, yes-no division served to identify interesting outliers for further investigation. The four classes (GR_{high}/SCS_{high} , GR_{low}/SCS_{high} , GR_{high}/SCS_{low} , and GR_{low}/SCS_{low}) are not intended to correspond to clearly delineated groups with a specific underlying biological meaning. Instead, we are using this classification as a simple descriptor of the data, akin to how FACS plots are often separated out into four groups. The cutoffs between low and high are chosen arbitrarily to facilitate description and discussion of the data. We make no claim of rigor for the division nor do we suggest this is an optimal division of the underlying distributions. We also suggest that one of the four classes so defined, phenotypic response with no consistent change in transcript profile, is likely to be a measurement artifact. However, this analysis does identify multiple outlying drug-cell line pairs in which strong transcriptional responses (and thus, drug activity in cells), is not matched by a change in phenotype. We therefore believe that the classification we present is both conceptually and practically useful.

1.12 Also on these two pages is a discussion of variability of responses. Again I find this disappointingly superficial. While no-one can argue with the fact that standard deviations (SDs) are legitimate measures of variability, they are hardly biologically meaningful, depending as they do on the assay. Also, empirical SDs have variances that depend strongly on their means, and so it is generally advised that any statistical analysis using SDs should be based on their logs. Further, it is not clear how many observations went into each estimated standard deviation. The caption to Fig 3b suggests 6, and if this is true, then these are very noisy SDs indeed. Finally, the provision of *p*-values for comparisons of collections of SDs, is an example of what I call *p*-value overkill. No null hypothesis is stated, but if it were, it would be that the distribution of some class of SDs is identical to that of some other class. Spelling it out like this highlights just how far this analysis is from anything that has any biological meaning. My reading of this aspect of the analysis is that the authors looked informally at the data, saw some trends, and decided to make their exploratory observations look scientific by decorating them with *p*-values. If so, this is in the best tradition of data snooping. If not, and the authors claim to be testing a hypothesis stated in advance of collecting the data, then that should be stated clearly and its scientific meaning explained.

R1.12 We have modified the analysis and figures in response to this criticism and re-written the text to ensure that our analysis is not misleading to the reader. Specifically, on the issue of *p*-values, we have re-computed the data and now illustrate variability across cell lines as a coefficient of variation for the SCS (the consistency of the transcript signature) and standard deviation for the GR_{AOC}. We use standard deviation because it is common in the biological literature even with *n*=6; we nonetheless appreciate that, from a statistical perspective, it would be preferable in this, and virtually all other molecular biology research, if *n* were higher. The issue, of course, is cost and time.

More importantly, the purpose of this figure was to compare variability across drug classes across two different types of data. This is a highly relevant question, because variability in the case of GR values corresponds to drug resistance and sensitivity, precisely the biological phenomenon whose origins we are attempting to understand with this sort of study. We wondered whether some classes of drugs were associated with a greater variation in response and the answer was clearly yes – whereas all cell lines assayed were sensitivity to cell cycle kinases inhibitors, their sensitivity to receptor tyrosine kinase inhibitors was very variable. The natural question is whether this variability is also manifest at the molecular (transcriptional level) and Fig. 3 suggests that the answer is yes. Since we are comparing variance within each class of measurement, not across different types of measurements (except in relative terms), we do not believe that the reviewer's concerns about the "meaning" of SDs is pertinent.

We disagree strongly with the reviewer's assertion that questions being addressed in this and other figures are not biologically meaningful and were based on "*exploratory observations [made] to look scientific by decorating them with p-values.*" The precise opposite is the case: we asked what any pharmacologist would consider the most obvious questions about the data and then attempted to find a way to answer the question in as rigorous manner as possible. We accept that other and perhaps better statistics could be used (and we made substantial changes in response to the reviewer's criticism), but biological irrelevance is most definitely not the issue.

With respect to the issue of "data snooping" we have attempted to be forthright that this paper has three components: (i) a report of a relatively large scale dataset of a novel type, collected to the best of our ability to do such experiments after a few years of optimization, (ii) a systematic analysis of the data as a means to identify key features and trends in a relatively unbiased manner – analysis that we believe to be improved thanks to the reviewer's suggestions, and (iii) attempts to get at the underlying biology by asking fundamental questions of the data and by investigating a specific use-case in detail. In the latter case, we use a different set of assays and wholly independent data, so the validity of our conclusions is not affected in any way by data "snooping."

The use case in Figs. 5-6 (and first introduced in Fig 4) is based on a biological conundrum we found inherently interesting – a set of drugs and cell lines in which a substantial change in transcription (indicative of drug action on its target) is not associated with a phenotypic effect. VIPER-based inference showed that FOXO3a was among the transcription factors whose activity was most dramatically changed by treatment of cells with MAPK, PI3K/mTOR and ErbB transcription factors. It therefore seemed worthy of attention from an unbiased perspective,

but literature knowledge also tells us that FOXO3a plays a critical role in cell proliferation and death. Our follow-on experiments (Figure 4) confirm these changes and suggest a role for FOXO3a in incomplete drug response. We believe that this type of analysis helps to demonstrate the potential value of large-scale response profiling, but we do not assert that this is the only biologically significant result that can be found in the data; instead, the example is meant to motivate others to pursue similar “mechanistic” explanations. We have changed the text to clarify this point.

1.13 Line 152. Here we meet GRAUC. In ref 16 by four of the present authors, it is explained that GRAUC has been superceded by GRAOC. Is this a typo, and AUC here should be AOC, or is the old AUC notion back in favour?

R1.13 Thanks – we have corrected this oversight and now use GR_{AOC} consistently. This is a case in which two efforts in the group evolved in parallel.

1.14 Line 194. Regarding the use of EOB, has there been any rationale given for the use of this particular definition for GR? Multiplicative definitions of independence are notoriously dependent on the “scale”, i.e. the nature of the measure. The use and misuse of measures of synergy has a large literature, but to my knowledge, none of it mentions GR.

R1.14 The reviewer is correct that synergy is a much misused concept whose calculation is fraught with complication. We have no particular love for the current definitions, but we find ourselves criticized in both directions by skeptics (among which we count ourselves) and enthusiasts. Because EOB (excess over Bliss independence) it is still commonly used as a measure for quantifying synergy, we have stuck with it (but since we provide all the numerical data, informed readers can substitute their own measures).

The reviewer is also correct that EOB has not been used previously in conjunction with GR. We now make clear in the text and methods that the EOB_{GR} has an analogous definition to EOB. While formally testing the EOB_{GR} as a universal metric of synergy would exceed the scope of this work, we include as a supplemental figure the classical EOB coefficients for the same experimental results. In all cases, we see synergy with both metrics. We have updated Figure 5 and Supplemental Figures 4 and 5.

To demonstrate further that the drug interaction we observed (“synergy”) is likely to be biologically informative, we now also show the GR values for a selected set of concentration for single and dual drugging, which also clearly demonstrates a clear qualitative switch from partial cytostasis to outright cytotoxicity.

1.15 Line 442. Fig. 6. The caption does not tell us what the grey lines are, although we might infer that they represent untreated control cells. Why are there replicate grey lines but not for the drug treated cells?

R1.15 The gray lines are indeed controls and we now state so in the figure legend. For the controls, we now show a single line representing the averaged behavior, as was done for the treated samples (the different graphical treatment was an oversight)

1.16 Lines 471 and 475. First, note that the captions of Supp Figs 5 and 6 have been switched. Second, although “untreated control cells” are mentioned in the caption to Supp. Fig 6(which should be 5), their color (grey) is not mentioned.

R1.16 We corrected the switched legends and included an explanation for the gray line in the figure legends.

1.17 Supp. Fig 6 seems rather poorly drawn.

R1.17 Thanks for pointing this out, we believe that the problem arose in the submission process. We will ensure that at the point of final submission, all vector images are of adequate quality for publication.

Reviewer #2 (Remarks to the Author):

2.1 *The data set is interesting, and showing transcriptional and phenotypic responses at this level is novel. Identification of drug synergies is also very important, However, the authors are not very clear on how this dataset can be exploited to identify potential synergistic drug partners for a given cell line.*

R2.1 The primary purpose of the current dataset was to compare measures of drug response at a molecular (transcript) and phenotypic level and to see how these measures correlate. Prediction of synergy was not a primary purpose of the study, although it does motivate the use case in Fig 5-6. We agree with the reviewer that there is as yet no general way to mine this type of data for insight into potentially synergistic drug pairs – we only made a suggestion that drug with transcriptional but no phenotypic responses are of higher interest. We hope that the availability of data such as ours will motivate the development of such tools. We have modified the text to make this clear and have also worked a lot on the on-line tool.

2.2 *Regarding the presentation: The paper is hard to read. Specifically the methodology is hard to follow if one is not completely familiar with the characteristic direction approach, for example. The manuscript will greatly benefit from a small description of the intuition behind this approach. There also seems to be a disconnect between the first half of the paper where the data is presented, described and high-level conclusions are drawn and the second half where the authors explore synergistic relationships. More clear descriptions of how one gets from the clustering to the identification of synergies will be very useful. Specifically, it is unclear whether the synergy described in BT-20 (MAPK and PI3K) is extracted from the data or from the literature.*

R2.2 By way of clarification: We apologize that the methodology was difficult to follow. We are a bit challenged by the concise format of this manuscript, but we have attempted to re-write the text to improve clarity. Specifically, we have re-written the methods section and attempted to provide intuitive explanations of the characteristic direction analysis and the clustering that gives rise to in Figure 1.

We also rewrote the section linking the large-scale analysis with the example of the synergy in BT-20 cells to make clear that the data in the current manuscript (and not the literature) are the basis for the hypothesis we test experimentally in Fig 5-6. We further point out that one of the uses of this data is to identify equivalence classes, which enables us to substitute drugs, possibly with more specificity and less toxicity, while maintaining the synergy.

2.3 *A technical description of the L1000 measurement system is lacking, and the authors refer to another publication which is in preparation (Line 343: “A full description of the L1000 method is being prepared for publication ...”). While we appreciate that all results can not be presented in a single manuscript, it is appropriate to share some of the data relevant to this work from the L1000 manuscript. Specifically, it is relevant to this work to show 1) how well the L1000 captures transcriptional profiles (how much we miss of the transcriptional responses by not doing a full transcriptome, and how L1000 compares to conventional transcriptional profiling on the genes measured by L1000) and 2) what is the reproducibility of the L1000 platform. There is some mention of replicate measurements and how their characteristic directions compare to random perturbations, but more detail is required (see Comment 6.2).*

R2.3 We appreciate this concern, which is being addressed through the preparation of a separate manuscript (with a very different set of authors) in which the L1000 approach is described in detail. We commit to making this paper available on the [biorxiv.org](https://www.biorxiv.org) preprint server prior to final acceptance of the current manuscript. Furthermore, we have substantially improved the description of the method and its reproducibility in a detailed protocol available on the support.lincscloud.org website.

2.4 *Similarly, for the data processing, there is insufficient information; for example, what is the “QNORM level 3” data? (p 15 at bottom.) How was this calculated from the intensity measurements? For the calculation of the significance of response in gene expression (p 16 at the top) a more detailed description should be given of how*

the empirical p-values were calculated (preferably with equations). For example, with how many random pairs were used to construct the null model?

R2.4 As we explained in our replies to reviewer 1, extensive changes have been made to the manuscript to address this concern (e.g. see R1.4).

The terms “QNORM” and “level 3 data” are explained in detail on the support.lincscloud.org website and we have added an appropriate reference. We would be happy to repeat these explanations in the supplementary material to the current paper but do not want to load the current manuscript down with previously published information; we propose to follow the editor’s advice on this point.

2.5 *Regarding the clustering approach: What was the reason for using fuzzy clustering? Did the fuzzy clustering provide a benefit over other clustering methods? It appears that in the end, the drug/cell line pairs are assigned to a single cluster again. Specifically, the authors state on p 16, L363: “...perturbations were assigned to the cluster in which they were found most often”. How does this relate to fuzzy clustering. Isn’t the point of fuzzy clustering that cluster membership is not discrete and that a perturbation can belong to multiple clusters?*

R2.5 We have substantially expanded the method section to describe each step of the clustering approach in detail and explain why a fuzzy clustering approach was taken. Briefly, using a fuzzy clustering algorithm strongly improve the reproducibility of the results across independent clustering runs which is a common observation for large, high-dimensional data.

2.6 *For the clustering analysis, a stability analysis is required before strong conclusions can be drawn from it. How does the clustering hold up under bootstrapping, for example? The authors should also provide a justification for the choice of k=20 and the fuzziness parameter of 1.22.*

R2.6 We have performed additional analysis and now include the benchmarking results for the clustering shown in Fig 1 in the Method section of the manuscript and in Supplemental Fig. 8; this justifies the clustering parameters used.

2.7 *For the synergy prediction, the specific case of synergy that is studied is one that is already well-known. Indeed the authors are aware of this, as they mention that “breast cancer cells typically requires both MAPK and PI3K signaling”. Thus, the finding that BT-20 cells depend on both MAPK and PI3K signaling is not novel. The authors should mention (also in the abstract) that they uncover a known synergy. As such it does still provide a validation of the strategy.*

R2.7 Initially, we had much the same reaction as the reviewer, but have discovered that *MAPK and PI3K* interaction is in fact considerably more complex and interesting than the reviewer’s comments might imply. Inhibitors of *MAPK and PI3K* signaling are synergistic (or even additive) only in a subset of breast cancer cell lines. In other cases, there is little or no evidence of drug interaction. Our analysis of BT-20 cells suggests that the operation of adaptive or compensatory responses between *MAPK* and *PI3K* activity is specific to some cell lines. We believe that this is the explanation for synergy, not simple “synthetic lethality”, as has been suggested previously. This extends what is known about such interactions and it allows us to identify equivalence classes in which different types of drugs can be substituted for each other as a means of achieving similar degrees of cytotoxicity in a combination. Overall, these represent substantial advances on current knowledge, while also serving to highlight a biological application of our data set.

2.8 *It was also not entirely clear how the authors arrived at this synergy prediction; was that entirely data-driven? It would be interesting if the authors could provide a list of the other cases of potential synergy that can be identified from the data in the same manner. Are there any novel predictions? It appears one other drug combination was tested and validated as well (Foretinib + A443654), but this is not further discussed.*

R2.8 We address these concerns with additional analysis, and a new supplementary figure.

The synergy finding was indeed data-driven and the text has been modified to make this clear (see R2.7). We chose to examine drug responses where the transcriptional and phenotypic responses were incongruent. In addition to the Erk-Akt synergy, we also identified other potential drug synergies, such as one between Foretinib and A443654 in Hs 578T cells. We found that the cell line Hs 578T, like BT-20, often shows limited phenotypic responses even when the same drugs elicit relatively strong transcriptional responses; these data are added to Fig. S4b. As the reviewer will no doubt appreciate (and as mentioned by reviewer 1 – R1.14), drug synergy is a fraught topic and we hesitate to make a list of potential synergies in the absence of careful follow-on experiments, which are beyond the scope of the current paper. However, there are no dramatic new predictions of synergy that we are withholding from the manuscript.

2.9 *For the synergy measurements, are there indeed no replicates? Also, how strong is a 'Bliss excess' of ~0.2-0.4? The significance of synergy is certainly dependent on the noise in the measurements. For example, the uncertainty in the single drug response measurements will feed through into the computation of the expected Bliss score, i.e. this score has a certain confidence interval. If the excess score is within this interval one can not call it significant. The authors should provide some estimate of the uncertainty on the Bliss score and attach a p-value to the excess score.*

R2.9 In response, to these concerns we have clarified the presentation and added a new figure panel.

Synergy was assessed in technical triplicates and repeated independently three times (which we believe is a thorough analysis). We provide a p-value on the aggregate synergy across the biological replicates and show the results of the biological repeats as a supplemental figure. Furthermore, the strength of the synergy can be illustrated by the fact that for a number of drug combinations the phenotypic response changes from a cytostasis in the presence of a single drug to cell death with drug combinations. We illustrate this point by including an additional panel in Figure 4, which demonstrates this phenotypic switch.

2.10 *The title of the section: "EGFR/MAPK inhibitors drive synergy by blocking drug adaptation" is too strong a statement. Also in the abstract the authors states that "we [...] show that it blocks adaptive drug resistance". In our opinion this is not supported by the evidence. While the authors show that the combined inhibition of MAPK and EGFR/ERBB2 leads to different dynamics in FoxO3A, it is not shown that this effect drives the synergy. Correlation does not imply causation. Intervention studies should be done to be able to prove that this mechanism is responsible/underlying the synergy. Additionally, the synergistic effect (in altering FoxO3A dynamics) is stronger between EGFR and PI3K inhibition, compared to MAPK and PI3K inhibition, hinting that more than one effect is at play here.*

R2.10 We agree with the reviewer that correlation doesn't identify causal relationships and have revised this section of the paper accordingly. In the current case, however, there is ample literature support that shows FoxO3A to be a key regulator of proliferation, supporting the statement that the sustained activation of FoxO3A is likely to be closely related to the synergy observed. Furthermore, the effect on FoxO3A in itself is synergistic, as the MEK/ERK and ErbB inhibitors by themselves have no effect on FoxO3A, but clearly work to sustain FoxO3A translocation into the nucleus in combination. We nevertheless follow the reviewer's suggestion have reworded the section title and reworked the text.

We assume that the increased strength of the synergy with EGFR and PI3K inhibition is caused by the additional inhibition of pathways downstream of EGFR other than the MAPK pathway. While this by itself has little effect on proliferation, in conjunction with PI3K inhibition it leads to stronger phenotypic effects. It is unknown if this effect is also exerted via FoxO3a, but this fact is not necessary to use equivalence classes to find possible drug substitutes.

2.11 *It is unclear what the benefit is of inferring gene expression of unobserved genes from the expression of the landmark genes, and for which analyses this is used. Only for the gene set enrichments?*

R2.11 As mentioned above we now make it explicit in the text when we are working with the ~1000 measured genes or with the inferred profile of ~20,000 genes. The use for the inferred expression profiles is reserved for enrichment analysis and VIPER.

2.12 *Identification of FoxO3a is based on a method (VIPER) that is not published yet, and no description is given of this method. Was FoxO3a the only hit or the top hit. If not, how was it selected?*

R2.12 The VIPER method has been published while this manuscript was under review (PMID: 27322546). We now cite the relevant publication and explain the method in a more detail in the method section. FoxO3a was chosen for follow-up experiments because we found that it has the second highest average score across PI3K, mTOR, AKT, MAPK, and ErbB perturbations, making it a likely master regulator of the BT-20 response.

2.13 *page 7 line 157. Cells sensitive to a specific inhibitor generally expressed the targeted receptor. How general is this. It is only shown with selected examples. This point should be supported in a more unbiased way?*

R2.13 We agree with the reviewer that this is a potentially important and interesting point (also addressed in two of our previous papers PMID: 24065145 and PMID: 24655548) that would be useful in understanding drug response more generally. Unfortunately, we have not yet analyzed sufficient cell lines for receptor expression and drug response to be able to make a rigorous claim. We have clarified in the text that these are limited examples and do not, by themselves, prove any general principle. We are, however, attempting to collect this type of data right now.

Reviewer #3 (Remarks to the Author):

3.1 *Whereas the study is conceptually not novel and several similar drug profiling approaches have been performed after the Connectivity Map (Lamb et al, 2006) was published, the authors employ an interesting and cost-effective method for transcriptional profiling to generate large datasets. However, the details and benchmarking experiments of this method are not part of this particular manuscript, making it difficult to judge its performance and diminishing the impact of the manuscript.*

R3.1 As stated in R2.3, we commit to making a pre-print of a detailed technical report on the L1000 paper available on [biorxiv.org](https://www.biorxiv.org) prior to acceptance of the current manuscript.

With respect to related work, we are not aware of drug profiling studies following the Lamb et al paper that attempted to connect high-throughput transcriptional profiling with phenotypic responses as a means to explore cell line-specific drug sensitivity. The relationship between these two types of data, and the extent of variation in such relationships, is the purpose of the current work.

3.2 *The strength of the manuscript lies in the large dataset and its analysis that connects drug response to transcriptional changes across time, dose and cell lines. This really is rather unique dataset and worth publishing. However, I struggle to identify the key novel findings that would make it a broad interest paper. Those highlighted by the authors seem somewhat expected and trivial to me. I.e., inhibition of well-conserved processes that are not differentially mutated in cancer cell lines yield similar profiles, whereas targeting highly context dependent and cancer driving processes (e.g., signaling pathways, RTKs) are much more variable. The synergy aspect of the paper is interesting but not yet worked out to a sufficient depth to make a case that this is really an effective approach to identify clinically meaningful synergies.*

R3.2 We agree with the reviewer that one key aspect of this manuscript is the dissemination and initial description of a novel and complex dataset, which will likely find use through further analysis. However, we are not aware that there is systematic public data showing that the transcriptional and phenotypic responses to drugs targeting signaling pathways, for example drugs inhibiting the MAPK or PI3K/AKT pathway, are cell type-specific both at the transcriptional and phenotypic level and how these levels relate.

We agree with the reviewer that research of this type does not lead immediately to clinically meaningful synergies. Considering the difficulty in identifying clinically relevant drug treatment from a study in cell lines, we never claimed or expected this. We do believe that our approach has shown that careful analysis of transcriptional and phenotypic profiling can identify functional drug combinations specific to certain cell lines and provide mechanistic insights. In addition, this work shows how drugs fall into equivalence classes for specific cell lines, which might allow for substitutions of drugs in cases where they might, for example, reduce toxicity effects.

More generally, we consider this work to be a proof-of-principle for the approach of mining transcriptional and phenotypic data to gain insights into drug responses. Further analysis of this data is certainly required to find more use-cases and as part of this publication, we will ensure that all data can be easily visualized and accessed by other researchers as is mandated by the LINCS grant.

3.3 *The mention of 8000 combinations is misleading as it suggests 8000 unique drug-cell line combinations.*

R3.3 We clarified the text to explicitly state that the '8000 signatures' refer to samples across different cell lines, drugs, concentrations, and time points. See also R1.1

3.4 *To call a cell viability measurement "Cellular phenotype profiling" seems a stretch*

R3.4 We changed the text to reflect this comment.

3.5 *To me, the described increase in amplitude with increasing concentration is not obvious in Fig. S2.*

R3.5 The reviewer correctly notes that Fig S2 does not show the relationship between the amplitude of the characteristic direction vector and drug concentration. We apologize for this error and now report the Spearman's correlation coefficient between amplitude and drug concentration in the text.

3.6 *Similarly, the claim that 10uM signatures of RTK and ECM inhibitors co-cluster across all cell lines is not easily visible from Figs. 1 and S1.*

R3.6 We agree with the reviewer that this is not immediately obvious from Figures 1 and S1. We expanded the text to describe the statistical test showing the clusters enriched for high drug doses of RTK and ECM inhibitors. We have also extensively improved the companion website to allow users to search for specific drugs: <http://amp.pharm.mssm.edu/LJP/drugs>.

3.7 *The authors should include a plot showing the data described for cell type specific responses of kinase inhibitors (lines 158-166) and cytostasis of alpelisib + ErbB inhibitors (line 223).*

R3.7 As suggested, we have added panels to Supplemental Figure 2 that show the cell line-specific responses to the VEGFR and IFG-1R inhibitors mentioned in the text. We further added panels to figures 5 and S4 that clearly demonstrate the shift from cytostasis to cell death in the synergistic combinations.

3.8 *At what time point were the synergy experiments measured? This seems an important parameter to me, given that these are class IV drugs, i.e. showing a transcriptional but no growth response.*

R3.8 The phenotypic response was measured at 72 hours both for the large-scale profiling and the synergy study. We have clarified this point in the text.

3.9 *Do the grey lines in Figs. 6 and S5 represent untreated or DMSO controls? If yes, these should be labeled as such; if not, these need to be included. Maybe representative IF images could help illustrate these findings. Further, the inhibition of MAPK signaling in Fig. S5a is not obvious to me.*

R3.9 We apologize for the confusion. As mentioned in response to reviewer one, the gray lines are indeed controls and we now state so in the figure legend. For the controls, we now show a single line representing the averaged behavior, as was done for the treated samples. The inhibition of MAPK is indeed subtle and difficult to see, owing predominantly to the fact that ERK is only weakly phosphorylated in actively cycling BT-20 cells.

3.10 *Fig. 1 is difficult to read. Maybe larger or more different symbols for time points and circling/grouping related clusters would help.*

R3.10 We agree that it is very difficult to visualize multi-dimensional datasets spanning time, concentration, cell line, drug target, and clusters. In response, we have continued to improve a dynamic web-based data browser that presents an interactive version of Figure 1. We now direct the reader to this website in the figure legend – see <http://amp.pharm.mssm.edu/LJP/drugs>

The version of the figure in the manuscript is necessarily static, but we are happy to take any suggestions by reviewers and editor into account when producing the final figure.

3.11 *In general, whenever individual perturbations/drugs are mentioned in the text, these should be highlighted in the corresponding figures to allow the reader to follow the line of argumentation (e.g. barasertib etc. in S3, lapatinib etc. in 4c and 5b).*

R3.11 We have added arrows to Figure 5b and labeled them in the legends to call out specific perturbations relevant to the text in. We also included an additional Supplemental Data file containing the classification of all perturbations into classes I through IV to allow readers to verify the class designations called out in the text and further explore the data.

3.12 *Right panel in Fig. 2a (Alk inhibitor), AZD6244 synergies (Fig. S4a) and Suppl. Data 2 are not referred to in text.*

R3.12 Sorry about that – we have modified the text and figures to ensure that all items are referred to in the manuscript.

3.13 *Fig. S1: The authors claim that chaperone and cell cycle inhibitors are active across all cell lines. This is not directly visible from the plots presented. The authors could plot e.g. the fraction of cell lines by drug class to illustrate this better. They should also show cluster 21 in the last panel.*

R3.13 The claim is that chaperones and cell cycle inhibitors generally elicit the same response (transcriptionally and phenotypically) in different cell lines. Figure 1 and S1 illustrate that clusters for chaperones, for example, the expression profile clusters 12 and 13 are comprised exclusively of drugs targeting chaperones but all six cell lines. Equally, the second panel in Figure 2a shows that phenotypically all six cell lines respond equally to NVP-AUY922 targeting HSP90. Plotting the fraction of cell lines by drug class would not illustrate this point as all cell lines also respond, for example, to PI3K inhibitors—albeit with different transcriptional profiles and phenotypic strength. We further emphasized this important distinction in the text.

We omitted “cluster 21” from the bottom panel because it is not a cluster per se, but rather the collection of perturbations that did not fall reliably into any other cluster (see improved description in the methods section). This group contained such a large number of conditions (1114) that it would make the graph more difficult to read. However, we did include this group of perturbations in the plot, labelled it with a star, and added a ‘broken histogram’ to indicate the number. We list the number of conditions within this group in the figure legend.

3.14 *Lines 105ff.: Mentioning the cosine distance in this paragraph would facilitate understanding.*

R3.14 We altered the text to improve the clarity of this paragraph as suggested by the reviewer.

3.15 *Line 144: In my opinion, there are more than 6 clusters containing signatures for multiple cell lines. Lines 129/130: Clusters 5 and 8 comprise more than one cell line.*

R3.15 There are indeed more than six clusters with multiple cell lines. Clusters 5 and 8 are comprised of more than one cell line (they are dominated by two lines). We have corrected this mistake and altered the manuscript accordingly.

3.16 *As the VIPER method is not yet published, a slightly more detailed description would be helpful.*

R3.16 As stated above, the VIPER method has been published while this manuscript was under review (PMID: 27322546). We now cite the relevant publication which includes a detailed description of the method.

3.17 *Legends for Figs. S5 and S6 are swapped and somewhat unclear (chDir? which cell line is used?).*

R3.17 We have corrected the swapped legends and clarified the text.

3.18 *What do DOX concentration and batimastat treatment refer to (line 369)?*

R3.18 These treatments were erroneously mentioned in this manuscript and the methods section has now been corrected.

3.19 *Some authors are not mentioned in the author contribution/ conflict of interest section.*

R3.19 We now include all authors in the 'Competing Interest' and 'Author Contributions' sections.

3.20 *Missing labels in figures: y axis in S2, labels for classes in S3, legend for grey lines in 6 and S5*

R3.20 We updated the figures to include all missing labels.

3.21 *Typos and unclear phrasing: line 79 significant, line 83 which combination?, line 110 s with, line 126 response (wrong word), line 167 of missing, line 263 are similar, line 280 effective, line 314 involved, line 324 room temperature, line 402 do; Fig. S1 107 M and y-axis label (distribution)*

R3.21 Thanks! - We have corrected all of the typos, ambiguities, and errors identified by the reviewer.

Reviewers' comments:

Reviewer #2 (Remarks to the Author):

Response to remarks by the authors on reviewer comments

Common and cell-type specific responses to anti-cancer drugs revealed by high throughput transcript profiling, Niepel et al.

In general the ms is much more readable than before and many comments have been addressed. However, there are still a few important points that have to be addressed before the manuscript can be acceptable to this reviewer. These points mainly pertain to clarifying what is described and providing more complete evidence to support claims. These are outlined below.

Note: Remarks that were satisfactorily responded to are omitted from the list below. Otherwise the original comment is numbered (e.g. 2.1), the response of the reviewers is indicated with an 'R' prefix to the corresponding number (e.g. R1.2) and the response of the reviewer to the authors' response is indicated with an 'RR' prefix to the corresponding number (e.g. RR1.2)

Reviewer #2 (Remarks to the Author):

2.2 Regarding the presentation: The paper is hard to read. Specifically the methodology is hard to follow if one is not completely familiar with the characteristic direction approach, for example. The manuscript will greatly benefit from a small description of the intuition behind this approach. There also seems to be a disconnect between the first half of the paper where the data is presented, described and high-level conclusions are drawn and the second half where the authors explore synergistic relationships. More clear descriptions of how one gets from the clustering to the identification of synergies will be very useful. Specifically, it is unclear whether the synergy described in BT-20 (MAPK and PI3K) is extracted from the data or from the literature.

R2.2 By way of clarification: We apologize that the methodology was difficult to follow. We are a bit challenged by the concise format of this manuscript, but we have attempted to re-write the text to improve clarity. Specifically, we have re-written the methods section and attempted to provide intuitive explanations of the characteristic direction analysis and the clustering that gives rise to in Figure 1. We also rewrote the section linking the large-scale analysis with the example of the synergy in BT-20 cells to make clear that the data in the current manuscript (and not the literature) are the basis for the hypothesis we test experimentally in Fig 5-6. We further point out that one of the uses of this data is to identify equivalence classes, which enables us to substitute drugs, possibly with more specificity and less toxicity, while maintaining the synergy.

RR2.2. The effort of the authors to clarify the methodology is appreciated, specifically the introduction of the signature consistency score (SCS). Unfortunately, the SCS raises new important questions. Specifically, the authors state the following: '. . . we defined the 'signature consistency score' (SCS) for each CD signature by comparing the average cosine distances between the two or three technical replicates to the average cosine distance between CDs of an equivalent number of randomly picked treatments (different drug and concentration, but same cell line and same time point). The value of the SCS is equal to negative log₁₀ of the fraction of random CDs with an average angle smaller than the average angle between technical replicates.'

RR2.2.1 How exactly is the SCS determined? We would strongly advise the authors to define the 'null hypothesis' clearly and then use equations to describe the procedure followed to arrive at the SCS. Specifically, what precisely is meant with 'equivalent number'? In response to Reviewer 1, the cutoff has been changed to 1.3, which is simply the log₁₀ of 0.05, the p-value cutoff in the previous version - so, unless I miss something, little has changed in terms of modeling the effect size.

RR2.2.2 What is the (average) variation in cosine distance between replicates? This is important in order to interpret the effect sizes in Figure 2b.

RR2.2.3 The authors state: 'we included only transcriptional profiles with an SCS > 1.3 as a means to filter noisy responses.' How many profiles were removed? (This is important to get an impression of the noise levels in the data).

2.3 A technical description of the L1000 measurement system is lacking, and the authors refer to another publication which is in preparation (Line 343: "A full description of the L1000 method is being prepared for publication ..."). While we appreciate that all results can not be presented in a single manuscript, it is appropriate to share some of the data relevant to this work from the L1000 manuscript. Specifically, it is relevant to this work to show 1) how well the L1000 captures transcriptional profiles (how much we miss of the transcriptional responses by not doing a full transcriptome, and how L1000 compares to conventional transcriptional profiling on the genes measured by L1000) and 2) what is the reproducibility of the L1000 platform. There is some mention of replicate measurements and how their characteristic directions compare to random perturbations, but more detail is required (see Comment 6.2).

R2.3 We appreciate this concern, which is being addressed through the preparation of a separate manuscript (with a very different set of authors) in which the L1000 approach is described in detail. We commit to making this paper available on the biorxiv.org preprint server prior to final acceptance of the current manuscript. Furthermore, we have substantially improved the description of the method and its reproducibility in a detailed protocol available on the support.lincscloud.org website.

RR2.3 This is simply not satisfactory. We do not know what will be published in the ms the authors pledge to put on biorxiv and what the consequences for the results of this manuscript will be. We are not asking much: a short description needs to be included here for the two points we mention above: 1) accuracy in the reconstruction of the complete transcriptome based on the L1000 landmark genes and 2) reproducibility of the L1000 platform (partially related to RR2.2).

2.4 Similarly, for the data processing, there is insufficient information; for example, what is the "QNORM level 3" data? (p 15 at bottom.) How was this calculated from the intensity measurements? For the calculation of the significance of response in gene expression (p 16 at the top) a more detailed description should be given of how the empirical p-values were calculated (preferably with equations). For example, with how many random pairs were used to construct the null model?

R2.4 As we explained in our replies to reviewer 1, extensive changes have been made to the manuscript to address this concern (e.g. see R1.4). The terms "QNORM" and "level 3 data" are explained in detail on the support.lincscloud.org website and we have added an appropriate reference. We would be happy to repeat these explanations in the supplementary material to the current paper but do not want to load the current manuscript down with previously published information; we propose to follow the editor's advice on this point.

RR2.4 Reference 19, where 'L1000 QNORM' and 'level 3 data' are mentioned, references the CD method. In R2.4 the authors mention the link 'support.lincscloud.org', but that link has been deprecated - exactly the reason why we have a preference for a reference to a publication, rather than a website. Later on in the text, where level 4 data is mentioned, there is a link to a website where the description can be found (<http://www.lincsproject.org/LINCS/tools/workflows/find-the-best-place-to-obtain-the-lincs-l1000-data>) where a description of Level 3 is given: Quantile-normalized gene expression profiles of landmark genes and imputed transcripts (Q2NORM or INF).

The authors also refer the reader to a link where MATLAB scripts for all of these calculations are available on <https://github.com/sorgerlab/L1000chDir>. This link is not working. Again, we are not asking much, so rather than sending the reader around a number of uninformative and non-functional links, simply describe in a few sentences what the input data is and also indicate whether the data is quantile normalized prior to or after reconstruction of the full gene expression profile - I can imagine that that could make a difference.

2.9 For the synergy measurements, are there indeed no replicates? Also, how strong is a 'Bliss excess' of $\sim 0.2-0.4$? The significance of synergy is certainly dependent on the noise in the measurements. For example, the uncertainty in the single drug response measurements will feed through into the computation of the expected Bliss score, i.e. this score has a certain confidence interval. If the excess score is within this interval one can not call it significant. The authors should provide some estimate of the uncertainty on the Bliss score and attach a pvalue to the excess score.

R2.9 In response, to these concerns we have clarified the presentation and added a new figure panel. Synergy was assessed in technical triplicates and repeated independently three times (which we believe is a thorough analysis). We provide a p-value on the aggregate synergy across the biological replicates and show the results of the biological repeats as a supplemental figure. Furthermore, the strength of the synergy can be illustrated by the fact that for a number of drug combinations the phenotypic response changes from a cytostasis in the presence of a single drug to cell death with drug combinations. We illustrate this point by including an additional panel in Figure 4, which demonstrates this phenotypic switch.

RR2.9.1 In Figure 5 something went wrong with the labeling of the panels. The caption and references in the text should be corrected.

RR2.9.2 In the heatmaps in Figure 5 a single replicate is shown. Please show the average of the replicates, and show the individual replicates in the supplement.

2.12 Identification of FoxO3a is based on a method (VIPER) that is not published yet, and no description is given of this method. Was FoxO3a the only hit or the top hit. If not, how was it selected?

R2.12 The VIPER method has been published while this manuscript was under review (PMID: 27322546). We now cite the relevant publication and explain the method in a more detail in the method section. FoxO3a was chosen for follow-up experiments because we found that it has the second highest average score across PI3K, mTOR, AKT, MAPK, and ErbB perturbations, making it a likely master regulator of the BT-20 response.

RR2.12.1 What were the protein networks being used in VIPER? Were these tissue type specific networks, and if so which networks?

RR2.12.2 FoxO3a had the 2nd highest score. What was the highest scoring candidate and why was it not chosen?

2.13 page 7 line 157. Cells sensitive to a specific inhibitor generally expressed the targeted receptor. How general is this. It is only shown with selected examples. This point should be supported in a more unbiased way?

R2.13 We agree with the reviewer that this is a potentially important and interesting point (also addressed in two of our previous papers PMID: 24065145 and PMID: 24655548) that would be

useful in understanding drug response more generally. Unfortunately, we have not yet analyzed sufficient cell lines for receptor expression and drug response to be able to make a rigorous claim. We have clarified in the text that these are limited examples and do not, by themselves, prove any general principle. We are, however, attempting to collect this type of data right now.

RR2.13 The authors make an interesting point: that RTK expression is necessary but not sufficient for sensitive to the corresponding RTK inhibitor. I am convinced that it will strengthen the manuscript significantly if the data to support this statement is shown in more detail (in spite of previous publications of this group). I realize that there are not sufficient data points to perform a statistical analysis, but showing a plot of the GR50(AOC) vs the (max) expression of the targeted RTK will already be very insightful. Specifically it is stated that: 'For example, Hs 578T cells expressed high levels of PDGFR/VEGFR receptors and were sensitive to nintedanib and foretinib (inhibitors of these receptors) . . .' This could be misleading as MCF7 is at least or perhaps even more sensitive to this compound. It is therefore important to know if MCF7 also expresses one of the RTKs targeted by these drugs.

Response to reviewers comments for NCOMMS-16-14817A

Reviewer #2 (Remarks to the Author):

RR2.2. The effort of the authors to clarify the methodology is appreciated, specifically the introduction of the signature consistency score (SCS). Unfortunately, the SCS raises new important questions. Specifically, the authors state the following: ‘. . . we defined the ‘signature consistency score’ (SCS) for each CD signature by comparing the average cosine distances between the two or three technical replicates to the average cosine distance between CDs of an equivalent number of randomly picked treatments (different drug and concentration, but same cell line and same time point). The value of the SCS is equal to negative log10 of the fraction of random CDs with an average angle smaller than the average angle between technical replicates.’

RR2.2.1 How exactly is the SCS determined? We would strongly advise the authors to define the ‘null hypothesis’ clearly and then use equations to describe the procedure followed to arrive at the SCS. Specifically, what precisely is meant with ‘equivalent number’?

Thank you for this comment. In the revised version of the manuscript we have added many more details about the calculation of the CD and SCS. The null hypothesis in this case is that signatures are drawn from a distribution that resembles an isotropic distribution; deviation from this null is associated with a higher SCS. The actual distribution of L1000 signatures used in this test is not precisely isotropic, because gene expression values are not independent from each other, and we therefore determined the distribution empirically from the data. As noted below, the source code used to evaluate CD and SCS is now available online.

In brief, for each technical replicate of each drug-induced perturbation, the characteristic direction (CD) was evaluated by comparing its L1000 QNORM vector to vectors for DMSO-treated controls on the same plate. L1000 data is provided at five levels in the data processing pipeline:

- **Level 1:** Raw unprocessed flow cytometry data from Luminex (LXB)
- **Level 2:** Gene expression values per 1000 genes after deconvolution (GEX)
- **Level 3:** Quantile-normalized gene expression profiles of landmark genes and imputed transcripts (Q2NORM or INF)
- **Level 4:** Gene signatures computed using z-scores relative to the plate population as control (ZSPCINF) or relative to the plate vehicle control (ZSVCINF)
- **Level 5:** Differential gene expression signatures

The normalized values for landmark genes used in the current work correspond in L1000 datasets to “level 3a” data (<http://www.lincsproject.org/LINCS/tools/workflows/find-the-best-place-to-obtain-the-lincs-11000-data>). Characteristic direction signatures were calculated per batch. A batch is a group of experimental conditions measured at the same time point and cell-line but on multiple plates as described using the following notation:

- M , the number of experimental conditions.
- N , the number of control replicates.

- J , the number of plates.
- $X_{i,j}$, a vector of length 978 (the number of genes in the L1000 assay) representing the j^{th} replicate of the i^{th} experimental condition. Note that since the replicates of an experimental condition are measured on different plates, j also, typically, denotes the plate.
- $C_{j,k}$, a vector of length 978 representing the k^{th} control replicate on the j^{th} plate.

First, we calculated the CDs for each experimental condition J times, each time using a replicate $X_{i,j}$ and the controls from the same plate to obtain $D_{i,j} = f(C_j, X_{i,j})$, where f is the CD function and C_j is the control matrix $[c_{j,1}, c_{j,2} \dots c_{j,K}]$ for the plate. Then the final CD, D_i , for an experimental condition is:

$$D_i = \frac{\sum_j D_{i,j}}{\left| \frac{\sum_j D_{i,j}}{J} \right|}$$

To calculate a null distribution of appropriately matching characteristic directions we define S_i as the mean of the pair-wise cosine distances between $D_{i,j}$ of the i^{th} experimental condition:

$$S_i = 1 - \frac{\sum_{j'=j+1}^J \sum_{j=1}^J [\text{Cos}(D_{i,j}, D_{i,j'})]}{\binom{J}{2}}$$

To estimate the null distribution of S_i , we randomly drew J number of $D_{i,j}$ from the pool of $M \cdot J$ conditions and calculated their average cosine distance as S_n . We repeated the process for 10,000 times to obtain the null empirical distribution.

The **Signature Consistency Score** (SCS) is the negative log of the one-tail comparison (on the lower end) of S_i with the null distribution S_n . In this case, the null hypothesis is that the $M \cdot J D_{i,j}$ are drawn from a distribution that resembles the isotropic distribution. The distribution used in the test is not exactly the isotropic distribution but rather was empirically determined because gene expression values are not independent from each other.

RR2.2.1 (continued): In response to Reviewer 1, the cutoff has been changed to 1.3, which is simply the log10 of 0.05, the p-value cutoff in the previous version - so, unless I miss something, little has changed in terms of modeling the effect size.

We are sorry for the misunderstanding. We have ourselves been uncertain on how to capture effect size and, as currently constituted, the CD and SCS methods do not consider it. As compared to the moderated Z-score (MODZ) and other methods for analyzing gene expression, the CD/SCS method gives more weight to genes that move in the same direction across repeats. A gene that changes less but ‘moves’ together with a large group of other genes might be scored higher than a gene that changed more in overall magnitude (i.e. had greater effect size). The CD method

accomplishes this by identifying the linear hyperplane that best separates control samples from the treatment samples using linear discriminant analysis, and then uses the Normal to this hyperplane to define the direction of change in expression space for each gene. The CD method is more sensitive in identifying the ‘correct’ differentially expressed genes than popular alternative methods data (e.g. limma, DESeq, significant analysis of microarrays and the t-test) as determined using several benchmarking strategies applied to the real.

Because this is an important but rather technical issue, we have written up a study fully describing how CD performs relative to other methods for scoring relative gene expression (Duan et al. (2016) *L1000CDS2: LINCS L1000 characteristic direction signatures search engine*. NPJ Syst Biol Appl. 2016;2. pii: 16015. doi: 10.1038) available at <http://rdcu.be/teqI>.

RR2.2.2 What is the (average) variation in cosine distance between replicates? This is important in order to interpret the effect sizes in Figure 2b.

To address this point and make Figure 2b clearer, we have added to the plot the empirically determined distribution of randomly chosen signatures (dashed grey lines) resulting from exposure of cells to unrelated drugs. As described in the response to point R2.2.1 we no longer use effect size in our calculations. All pairs of signatures are scored on the basis of the angle between them.

It is worth noting that the distribution of variance between technical replicates itself varies with the cell line, time point, and treatment. This is why Figure 2b depicts only cosine distances between pairs of perturbation that have been judged to be significant base on an $SCS > 1.3$.

RR2.2.3 The authors state: ‘we included only transcriptional profiles with an $SCS > 1.3$ as a means to filter noisy responses.’ How many profiles were removed? (This is important to get an impression of the noise levels in the data).

Overall, we scored only 2864 of the 7825 (37%) of the tested conditions as significant by the $SCS > 1.3$ metric. We now report these values in the manuscript.

RR2.3 This is simply not satisfactory. We do not know what will be published in the ms the authors pledge to put on biorxiv and what the consequences for the results of this manuscript will be. We are not asking much: a short description needs to be included here for the two points we mention above: 1) accuracy in the reconstruction of the complete transcriptome based on the L1000 landmark genes and 2) reproducibility of the L1000 platform (partially related to RR2.2).

We apologize for this problem although much of what the reviewer is asking for has been available for some time on the LINCS website. We are now happy to report that a complete description of the L1000 method now available in Subramanian et al “A Next Generation Connectivity Map: L1000 Platform And The First 1,000,000 Profiles” (2017) at biorxiv.org (see:

<http://biorxiv.org/content/early/2017/05/10/136168>). The manuscript has also been submitted for peer review; if the current manuscript is accepted, we will update the citation for Subramanian et al.

RR2.4 Reference 19, where 'L1000 QNORM' and 'level 3 data' are mentioned, references the CD method. In R2.4 the authors mention the link 'support.lincscloud.org', but that link has been deprecated - exactly the reason why we have a preference for a reference to a publication, rather than a website. Later on in the text, where level 4 data is mentioned, there is a link to a website where the description can be found (<http://www.lincsproject.org/LINCS/tools/workflows/find-the-best-place-to-obtain-the-lincs-l1000-data>) where a description of Level 3 is given: Quantile-normalized gene expression profiles of landmark genes and imputed transcripts (Q2NORM or INF). The authors also refer the reader to a link where MATLAB scripts for all of these calculations are available on <https://github.com/sorgerlab/L1000chDir>. This link is not working. Again, we are not asking much, so rather than sending the reader around a number of uninformative and non-functional links, simply describe in a few sentences what the input data is and also indicate whether the data is quantile normalized prior to or after reconstruction of the full gene expression profile - I can imagine that that could make a difference.

Again, we apologize for these error. We are not sure why these links were deprecated. We believe that all of these concerns are mitigated by pre-publication release of the L1000 paper. Our paper still requires a link to the code of course, and apologize that we had not toggled the relevant section of our github repo to *public*. This is now fixed.

RR2.9.1 In Figure 5 something went wrong with the labeling of the panels. The caption and references in the text should be corrected.

We think that we have identified and corrected the issue in figure 5 raised by the reviewer. We will work with the editorial team to sort out any remaining problem.

RR2.9.2 In the heatmaps in Figure 5 a single replicate is shown. Please show the average of the replicates, and show the individual replicates in the supplement.

Showing all repeats of all two-way dose response curves is not a simple matter (at least not within the confines of a relatively brief paper). Having considered this request carefully we have decided to plot (Figure 5a) GR values for drugs individually and in combination at a single relevant concentration across all replicates to enable visual estimation of the error. To depict drug interaction at multiple concentrations, we have plotted a heatmap (figure 5b). It is not possible to show errorbars on a heatmap, and readers interested in seeing how drug interaction varies with the experiment will need to look at the primary data in a software tool such as MatLab (we provide the data of course). We are willing to adding a series of additional supplemental figures showing all replicates, but believe it makes more sense to direct interested readers to the primary data.

RR2.12.1 What were the protein networks being used in VIPER? Were these tissue type specific networks, and if so which networks?

The regulon used in this analysis was built on the TCGA data for breast cancer and contains all genes present in the L1000 inferred data as inputs and transcription factor genes as output. We have added a section in the Methods to describe this aspect of this analysis in detail.

RR2.12.2 FoxO3a had the 2nd highest score. What was the highest scoring candidate and why was it not chosen?

We have reworked this section of the manuscript to suggest that we chose FoxO3a on first principles (as a known target of Akt and Erk) and then confirmed that it had a high VIPER score. The VIPER algorithm provides rather long lists of possible transcriptional regulators, many of which have relatively limited annotations. Further investigation has shown that where FoxO3a lies on the VIPER list depends on settings in the algorithm and the precise set of data being considered (although FoxO3a almost always lay in the top five transcription factors). It seems safer to us to simply make clear that we made a decision to looking at FoxO3a because we had good reagents and it made sense.

We are nonetheless sympathetic with the thrust of this comment. It would be very helpful if high throughput methods such as VIPER returned a list of candidate regulators that could be examined systematically as a way to validate the computation. However, many of the genes at the top of the VIPER list (e.g. SOX13, NFE2L1, CREB3L4, ZC3H10, IRF7, or SUPT6H) are not ones for which we have tested reagents and sorting this out would take many months of effort in the best cases and be impractical in many others. Experiments of the type we describe are therefore limited by necessity. It is fair, in our opinion, to describe such experiments as having been guided by high throughput studies and as a demonstration of the value of such studies, but it cannot be claimed that they *validate* the high throughput results in any rigorous way since they do not systematically evaluate logically chosen sets of hypotheses. We have edited the text in multiple locations to make this point clear.

RR2.13 The authors make an interesting point: that RTK expression is necessary but not sufficient for sensitive to the corresponding RTK inhibitor. I am convinced that it will strengthen the manuscript significantly if the data to support this statement is shown in more detail (in spite of previous publications of this group). I realize that there are not sufficient data points to perform a statistical analysis, but showing a plot of the GR50(AOC) vs the (max) expression of the targeted RTK will already be very insightful. Specifically it is stated that: 'For example, Hs 578T cells expressed high levels of PDGFR/VEGFR receptors and were sensitive to nintedanib and foretinib (inhibitors of these receptors) . . .' This could be misleading as MCF7 is at least or perhaps even more sensitive to this compound. It is therefore important to know if MCF7 also expresses one of the RTKs targeted by these drugs.

We agree with the reviewer that there is interesting biology to be discovered in relating receptor expression, ligand response, and drug sensitivity to each other, but this probably requires an additional manuscript. We focus here on BT20 to make the point. Analysis of the other cell lines is descriptively complex since there are six lines (and thus a lot of plots), but as the reviewer points out this number is unfortunately not enough for a solid statistical analysis. Thus, the connection must be left as a speculation. We hope that the publication of this manuscript and the growth of the CMAP database will encourage other to investigate this direction.

With respect to the specific issue about MCF7 cells raised by the reviewer, it is important to recall that RTK inhibitors exhibit very substantial polypharmacology, especially in the case of drugs that target cMet, PDGFR, VEGFR, and FGFR receptors (we point this out when examining CD over drug dose). The discrepancy noted by the reviewer is not in fact a discrepancy, but rather a limitation of our knowledge about relevant drug targets. We have rephrased the manuscript to clarify our interpretations and more completely explain the limitation of the current analysis.

REVIEWERS' COMMENTS:

Reviewer #2 (Remarks to the Author):

In general the authors addressed all the comments and we advise accepting the manuscript for publication. There are, however, a number of nagging little practical problems (link that still does not work, erroneous figure legend) that should have been solved. We encourage the authors to take care of this as this makes the work accessible and ensures that this resource can be used, which is the goal of the authors, I would assume.

Here are some minor points:

RR2.2.1 How exactly is the SCS determined? We would strongly advise the authors to define the 'null hypothesis' clearly and then use equations to describe the procedure followed to arrive at the SCS. Specifically, what precisely is meant with 'equivalent number'?

We appreciate the mathematical description - this clarifies the process a great deal. The authors give a description of how the null distribution is computed, not what the null hypothesis is. We do not want to belabor this point, but it helps the reader to clearly state what you are testing here. For example, 'The null hypothesis is that the variation of the CD between replicates in the same cell line and treatment is equal to the variation between the CDs of an equal number of CDs randomly selected from different cell lines and treatments. If the replicates of the same treatment and cell line show a significantly smaller variation than the random samples, one can reject the null hypothesis'. Or something similar.

RR2.4 This link: <https://github.com/sorgerlab/L1000chDir> is still not operational.

RR2.9.2 the legend of Figure 5 is still incorrect.

Small: 'AS is a founder Genometry...' > 'AS is a founder of Genometry...'

Response to reviewers comments for NCOMMS-16-14817A

Reviewer #2 (Remarks to the Author):

In general the authors addressed all the comments and we advise accepting the manuscript for publication. There are, however, a number of nagging little practical problems (link that still does not work, erroneous figure legend) that should have been solved. We encourage the authors to take care of this as this makes the work accessible and ensures that this resource can be used, which is the goal of the authors, I would assume.

We thank the reviewer for his positive comments and careful review of our work. We are sorry for the last remaining issues that we have now corrected. We will take special to ensure that our work is accessible and as easy to reuse as possible.

Here are some minor points:

RR2.2.1 How exactly is the SCS determined? We would strongly advise the authors to define the 'null hypothesis' clearly and then use equations to describe the procedure followed to arrive at the SCS. Specifically, what precisely is meant with 'equivalent number'?

We appreciate the mathematical description - this clarifies the process a great deal. The authors give a description of how the null distribution is computed, not what the null hypothesis is. We do not want to belabor this point, but it helps the reader to clearly state what you are testing here. For example, 'The null hypothesis is that the variation of the CD between replicates in the same cell line and treatment is equal to the variation between the CDs of an equal number of CDs randomly selected from different cell lines and treatments. If the replicates of the same treatment and cell line show a significantly smaller variation than the random samples, one can reject the null hypothesis'. Or something similar.

We thank the reviewer for the good suggestion and we included a similar wording in the methods.

RR2.4 This link: <https://github.com/sorgerlab/L1000chDir> is still not operational.

The repo was private as it was hard to give access exclusively to the reviewers. We have now made the repo public and check that dependencies are properly described.

RR2.9.2 the legend of Figure 5 is still incorrect.

The legend has been corrected to reflect the different panels.

Small: 'AS is a founder Genometry...' > 'AS is a founder of Genometry...'

Thanks for pointing out the typo.